



# Isotopic signatures of methane emission from oil
# and natural gas plants in southwestern China
Dingxi Chen[1†], Yi Liu[2†], Zetong Niu[1], Ao Wang[1], Pius Otwil[1], Yuanyuan Huang[1],
Zhongcong Sun[1], Xiaobing Pang[3], Liyang Zhan[4], Longfei Yu[1*]
[1]Shenzhen Key Laboratory of Ecological Remediation and Carbon Sequestration,
Institute of Environment and Ecology, Tsinghua Shenzhen International Graduate
School, Tsinghua University, Shenzhen 518055, China
[2]Safety, Environment and Technology Supervision Research Institute of PetroChina
Southwest Oil and Gas Field Company, Chengdu 610041, China
[3]College of Environment, Zhejiang University of Technology, Hangzhou, 310014,
China
[4]key Laboratory of Global Change and Marine-Atmospheric Chemistry, Third
Institute of Oceanography, Ministry of Natural Resources, Xiamen, 361005, China
[*]Corresponding authors: longfei.yu@sz.tsinghua.edu.cn
[†]These authors contributed equally.



## Abstract

Methane ($CH_4$) emissions to atmosphere from Chinese oil and gas (ONG) sector are subject to considerable uncertainty. The isotopic composition of $CH_4$ isotopes ($\delta^{13}C$) varies between emission sources, enabling the identification of changes in specific $CH_4$ sources. However, there are few relevant studies in China, especially at the ONG site level. We obtained $CH_4$ mixing ratios and isotopes from atmospheric samples collected by UAV and ground monitoring, and employed the HYSPLIT model to investigate $CH_4$ distribution at ONG sites in southwest China. It was found that the $CH_4$ isotopic signatures provide a strong basis for the emission intensity at the ONG sites. The meteorological and site conditions were identified as the most influential factors in $CH_4$ distribution at sites. The $CH_4$ from the equipment area contributed approximately a quarter of the $CH_4$ observed over the sites. The $CH_4$ source isotopic signatures ($\delta^{13}C$) of this study were heavier than those globally, indicating that they were mainly thermogenic sources. Finally, the heavier $\delta^{13}C$ of this region may lead to an overestimation emission of global $CH_4$ from fossil fuel sources by 3.47 Tg $CH_4$ $yr^{-1}$, and underestimation from microbial sources. This study highlights the importance of regional $CH_4$ isotopes, with great significance for $CH_4$ inventories of global sectors.



## 1 Introduction

Methane ($CH_4$) is a major greenhouse gas (GHG) in the atmosphere, possessing a global warming potential 82.5 times greater than carbon dioxide ($CO_2$) over a 20-year timeline, and 29.8 times greater over a 100-year period(Ipcc, 2021). The mixing ratios of $CH_4$ in the atmosphere has increased by 150% since the industrial revolution, primarily driven by human activities(Hmiel et al., 2020; Saunois et al., 2016b; Tian et al., 2016; Skeie et al., 2023). The oil and natural gas (ONG) industry is one of the major contributors to anthropogenic $CH_4$ emissions accounting for approximately 25% of global emissions. Over the past 20 years, $CH_4$ emissions from the ONG industry have increased by about 23.1%, corresponding to an average annual growth rate of 1.1%(Lauvaux et al., 2022). However, some studies suggest that $CH_4$ emissions from fossil fuel sources are likely to be seriously underestimated(Lauvaux et al., 2022; Hmiel et al., 2020). In China, $CH_4$ emissions from ONG industry have been estimated to increase from 116.6 Gg in 1990 to 1124.8 Gg in 2018(Epa, 2019), but various nationwide investigations tend to be highly variable and uncertain(Zhang et al., 2014)ˌ(Sun et al., 2022). Such discrepancy primarily arises from the scarcity of publicly available data and the accuracy of emission factors. The emission factor (EF) data used mainly come from the IPCC and its improvements, which may not accurately reflect the actual situation in China(Gao et al., 2022).

Overall, the general consensus is that the global $CH_4$ mixing ratios have been increasing over the past few decades(Schwietzke et al., 2016; Montzka et al., 2011; National Oceanic & Atmospheric Administration, 2024a). At present, the main controversy is the contribution sources of $CH_4$ (the drivers of the atmospheric $CH_4$ growth) and the high uncertainty of contribution (the uncertainty in $CH_4$ budget)(Kirschke et al., 2013; Saunois et al., 2016a; Rice et al., 2016; Tibrewal et al., 2024). The identification of $CH_4$ sources is essential for the estimating and reduction of $CH_4$ emissions.

The anthropogenic sources $CH_4$ accounting for about 50-65% of global $CH_4$ emissions come from human activities, including ONG industry, wetlands, agriculture



(e.g., ruminants, rice cultivation), landfills, and wastewater. However, the contribution
of each $CH_4$ source is highly uncertain(Skeie et al., 2023). The presence of greenhouse
gases is attributed to a multitude of sources. These sources exhibit distinct isotopic
signatures, which serve as a valuable tool for the identification and differentiation of
their origins(Schwietzke et al., 2016). Stable isotope is one of the common tools to
distinguish different sources of the same substance(Suzuki, 2021; Peng et al., 2024;
Leitner et al., 2020; Basu et al., 2022). A number of isotope pool mixing-models have
been developed to quantify source contributions(Parnell et al., 2013)·(Barthold et al.,
2011), providing important constraints on the role of various anthropogenic and natural
emissions to the overall greenhouse gas burden, and thereby enhancing our
understanding of the complex dynamics of climate change(Zhang et al., 2022; Rigby et
al., 2017). Several studies have attempted to use $CH_4$ isotopes for unraveling regional
$CH_4$ emission patterns. Using the characteristics of carbon ($\delta^{13}C$) and hydrogen ($\delta D$)
stable isotopic of $CH_4$, help distinguish between specific emitters of $CH_4$ from the
Condamine region, Queensland, Australia (main $CH_4$ sources include coal seam gas
related, piggery, ground and river seeps, feedlot and grazing cattle, landfill and others),
the $\delta^{13}C$ and $\delta D$ signatures of each $CH_4$ source were analyzed(Lu et al., 2021). Some
researchers combined $\delta^{13}C$ with other models to separate industrial $CH_4$ emission
sources from atmospheric(Assan et al., 2018). A recent research indicated that 85% $CH_4$
emissions growth from microbial sources during the period 2007 to 2016 were
estimated based on $\delta^{13}C$ of $CH_4$ in the atmosphere(Basu et al., 2022). A study based on
atmospheric $\delta^{13}C$ of $CH_4$ data showed that $CH_4$ emissions from the fossil fuel sector
remained largely unchanged at the 1980s and 1990s levels, but increased significantly
between 2000 and 2009(Rice et al., 2016). By analyzing $\delta^{13}C$ of $CH_4$, researchers
suggested that a reduction in microbial $CH_4$ emissions in the Northern Hemisphere may
have contributed to the stabilization of atmospheric $CH_4$ over the Millennium(Kai et al.,

2011).

Global observations and researches on $CH_4$ source isotopic signatures from the ONG
industry have been carried out and some results obtained, such as a global database of
$CH_4$ isotopes from fossil fuels in the atmosphere has been established, according to the



latest research results to update timely(Schwietzke et al., 2016). Other studies have been
initiated at regional and urban scales, such as estimating $CH_4$ emissions from
abandoned ONG wells in the United States, and $\delta^{13}C$ of $CH_4$ has been employed to
distinguish the coalbed and nature gas sources(Townsend-Small et al., 2016). Another
research also reported the $CH_4$ isotopic signatures of ONG fields in Romania,
confirmed $CH_4$ in the region mainly from the ONG sector, and simultaneous resulted a
wide range of $\delta^{13}C$ values, which indicated regional variation in $CH_4$ isotopes(Menoud
et al., 2022). Recent advancements in UAV technology have facilitated novel
approaches to monitor and quantify $CH_4$ emissions, particularly in localized
settings(Shaw et al., 2021). For example, the airborne platform was used to monitor the
$CH_4$ emission from UK and Dutch offshore ONG installations, to quantify and identify
the sources(France et al., 2021), and UAV-based sampling systems were used to analyze
greenhouse gas stable isotope(Leitner et al., 2023). Nevertheless, the relevant studies
were largely concentrated in foreign countries, and a few reports have revealed the $CH_4$
isotopic values from Chinese ONG production regions (SI, Table S1). To date, no
research has examined the site-specific isotopic signatures of $CH_4$ emissions within the
industrial site in China. In addition, the $CH_4$ isotopic signatures at the site level is
regulated by many factors, such as source types(Zhang and Zhu, 2008; Schoell, 1980;
Liu et al., 2019), processing (e.g., purification or production of light hydrocarbon),
meteorological condition, sampling method, size of the site and so on. Therefore, the
isotope tools are likely to provide quantitative or semi-quantitative reference for
investigating regional or site-level $CH_4$ emission hotspots.

In order to fulfill for the lack of site-level $CH_4$ emission research, this study aims to

delineate the isotopic traits of $CH_4$ emitted from Chinese ONG industry stations, by
analyzing the sources of $CH_4$, and judging whether there is $CH_4$ emission in the field
stations combined with other information. Simultaneously, the database of $CH_4$ isotopes
in China will be enriched. We conducted monitoring and sampling of $CH_4$ at 11 ONG
sites in the central Sichuan Basin, China. We characterized the sources and isotopic
signatures of $CH_4$ based on isotope data derived from these samples. This study's data
significantly augments the $CH_4$ isotope database for ONG source in the central Sichuan



Basin, effectively bridging the previously existing gaps in both surface and upper
atmospheric $CH_4$ isotope data for these sites. Looking ahead, this expanded dataset will
serve as a foundational resource for future research, enabling more comprehensive
assessments of global $CH_4$ emissions and their sources, and potentially guiding the
development of targeted mitigation strategies for the ONG industry.



## 2 Method

**2.1 Study sites**

The study area is located in Sichuan Basin, Southwest China, where about 19 % of the country's total natural gas reserves have been discovered(The People's Government of Sichuan Province, 2024). Until 2022, the region has about 77,000 km gas pipelines(National Bureau of Statistics, 2024). Between 2013 and 2023, natural gas production in this region increased from $21.31 \times 10^9$ to $59.48 \times 10^9$ $m^3$(Sichuan Provincial Bureau of Statistics, 2024), with an average annual growth rate of about 11%. In 2020, ONG production in Sichuan accounted for 24% of China's total ONG production (National Bureau of Statistics, 2024). We monitored $CH_4$ mixing ratios and sampled air for isotope measurements across 11 ONG processing or transportation stations in the central Sichuan Basin. The study region is characterized by a humid subtropical climate, with consistently warm and humid conditions throughout the year. The areas of these stations vary from 2,000 to 300,000 $m^2$, while the production activities also vary, including natural gas purification plants, gas gathering stations, light hydrocarbon plants, pigging stations, pressurization stations, etc. (Table 1). Most of the ONG stations are located in rural areas, surrounded by mountains, forests, farmlands, and reservoirs, and there are rivers located in the vicinity of only several sites. Paddy fields are the main farmland in this area, and rice is the main grain crop(Sichuan Provincial Department of Agriculture and Rural Affairs, 2024). For reasons of privacy and confidentiality, the specific locations and contours of the ONG stations cannot be disclosed in this paper.



**Table 1 Background information of the studied production/processing sites for oil and natural gas**

| Site | Type | Area (m²) | Processing capacity ($10^4$ m³/d) | Surrounding environment | Activity |
|---|---|---|---|---|---|
| MX (S1) | Purification plant | 25000 | 445 | Forests, farmland | Natural gas processing, including membrane separation, adsorption, desulfurization, dehydration and other processes |
| SN (S2) | | 113000 | 3000 | Forests, farmland, many ponds | |
| DQ (S5) | Gas gathering stations | 5096 | 115 | Forests, reservoirs | Gas Collection and transportation |
| XBQ (S3) | | 9420 | 278 | Forests, farmland | |
| XQ (S4) | | 4220 | 1000 | Farmland, ponds | |
| QTCSN (S7) | Light hydrocarbon plant | 6650 | 10 | Forests, farmland | $C^{3+}$ component of natural gas was recovered by low temperature separation process |
| QTCSZ (S8) | | 25257 | 30 | Forests | |
| LHZ (S9) | Union Station | 7958 | 2700 | River | ONG centralized treatment, sewage treatment, product output |
| ZYZ (S11) | Supercharging station | 7740 | 24 | Farmland, ponds | Pressure and transmission |
| L1 (S10) | Central well station | 8679 | 90.3 | Forests, farmland, ponds | Gas Collection and transportation |
| XM (S6) | Pigging station | 5167 | 630 | River | |

## 2.2 Sampling methods

From 13 April to 19 April 2023, we monitored and collected samples at 11 ONG production stations in the central Sichuan Basin, obtaining a total of 74 samples, including 28 ground samples and 46 air samples. Ground samples were collected at heights ranging from -0.5 m (0.5 m under the ground) to 2 m. Sampling locations were



chosen in open areas of each station, including areas near pipelines and production
equipment. Sampling in the area of pipelines and production equipment was performed
at locations that show abnormal mixing ratios after ground monitoring, and in instances
where no apparent $CH_4$ emission was detected, sampling was performed in the center
of the pipelines and production equipment area. For large field stations (> 10,000 $m^2$),
multiple sampling points were established, while for small field stations (< 10,000 $m^2$),
1-2 sampling points were established in facility areas, and the sampling time for each
sample was about 45-50 seconds. The drone was launched in open positions. Air
sampling was performed by an Unmanned Aerial Vehicle (UAV) equipped with an
automatic sampling pump (Fig. 1). The UAV model was a DJI-T10 upgraded version,
and the sampling pump model was KVP04-1.1-12V (1.25 L/min). Taking into account
the altitude ranges utilized in previous studies(Kim et al., 2025; Han et al., 2024; Chen
et al., 2024; Liu et al., 2021; Liu, 2018; Ali et al., 2017), along with the drone's flight
endurance and sampling duration, the monitoring altitude for this study is defined,
sampling heights were 50 m, 100 m, 200 m and 300 m respectively. Initially, a ground
sampling site was identified, typically within the pipeline vicinity of the plant.
Subsequently, a UAV equipped with an automatic sampling pump and air collection
bags was lifted to altitude of 300 m above the ground sampling site. The UAV then
sequentially descended to altitudes of 200, 100, and 50 m, respectively, dedicating 45
to 60 seconds at each elevation for collecting air samples. This systematic approach
ensures a comprehensive and stratified sampling strategy, facilitating the assessment of
atmospheric constituents at varying heights. The volume of each air sample was
approximately 1 L. All sites sampled at altitude, with the exception of S1, which
sampled at 200 m and 300 m, all other stations sampled at various altitudes. Air
sampling and UAV cruising were synchronized. HOONPO Teflon gas bags (1L, 2L
specifications) were used for gas sampling. In addition, we sampled the region near an
urban park and river to analyze the region's atmospheric $CH_4$ mixing ratio and isotope
information, collecting a total of 4 samples (2 from the riverbank, 2 from the park). We
also sampled a production well (built in the 1980s) that was out of repair and had
significant emissions, and the result was used as a reference for source signal analysis,



with a total of 3 samples collected (one from the open area of the site and two from the
leak). The sample list and test results are provided in (SI, Table S2).

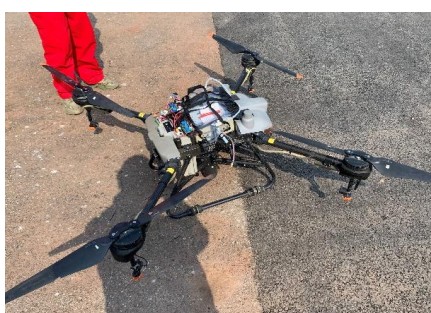
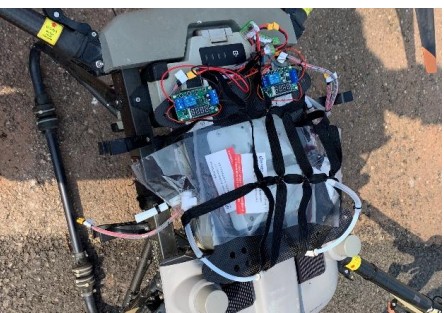


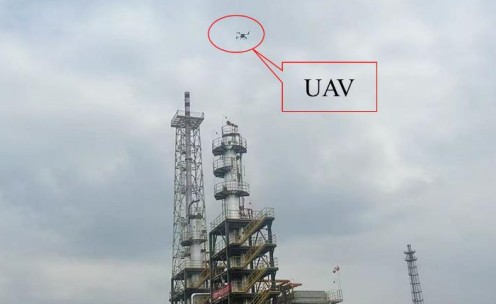


**Fig. 1** UAV, automatic sampling system and sampling over the site.

The influence of meteorological conditions on the CH$_4$ mixing ratio and isotopes at
the field station was also considered. Therefore, a portable meteorological station was
deployed at each station during the sampling or monitoring periods. It was equipped
with a three-dimensional ultrasonic wind speed and direction sensor (model: M307200),
which recorded the wind speed (horizontal and vertical) and direction (horizontal and
vertical) near the ground (3 to 10 m according to field conditions), the sampling
frequency is 32 Hz with a resolution of 0.1 m/s for wind speed and 0.1° for wind
direction, and the accuracy of wind direction and speed is 2° and 0.2 m/s, respectively.
We also obtained air pressure, solar radiation, temperature, and relative humidity from
weather stations. Since the meteorological conditions at high altitudes (50 to 300 m)
cannot be monitored by instruments, we used the HYSPLIT model to test the influence
of wind speed and direction on our measurements of CH$_4$ mixing ratios and isotopes at



high altitudes.
**2.3 Measurement methods**
Gas samples were analyzed within one month after on-site sampling. Picarro G2132-
i was used to detect the isotope and mixing ratio of $CH_4$, which is based on unique
Cavity Ring-Down Spectroscopy (CRDS). The $\delta^{13}C$ detection accuracy (1-σ, 1-hour
window) of the instrument is as follows: when the mixing ratio of $CH_4$ is greater than
1.8 ppm, the accuracy of 5-minute mean value is less than 0.8‰, when the mixing ratio
of $CH_4$ exceeds 10 ppm, the accuracy is less than 0.4‰(Picarro, 2024). The calibration
of the instrument was performed with $CH_4$ isotope standard gas (2.8 ppm, -68.6±0.3‰),
produced by Airgas company, USA. Standard gas measurements were performed daily,
before and after the start of the sample test, to correct the same-day test data (the
correction parameters of $CH_4$ isotope and mixing ratio were approximately 1.5‰ and
0 ppm, respectively). Each gas bag sample underwent three repeatedly measurements,
totaling 74 samples. For each sample measurement, analysis over 180 seconds was
performed on the Picarro G2132-i CRDS, and the average of the last 120 seconds of
$CH_4$ isotope and mixing ratio data was recorded as the sample assay value. Both $CH_4$
isotope and mixing ratio data are available for each test.

**2.4 Calculation of source isotopic signatures**
Based on the sample detection data, the method of Keeling plot method was used to
determine the $CH_4$ source(Keeling, 1958; Pataki et al., 2003) for each field station, as
shown in formula (1):

$\delta_{(a)}=[CH_{4(b)}]\ (\delta_{(b)}-\delta_{(s)})\ \cdot 1/[CH_{4(a)}]+\delta_{(s)}$                 (1)

Where $\delta_{(a)}$, $\delta_{(b)}$, and $\delta_{(s)}$ represent the $\delta^{13}C$ values of the sample, the background air
and the average source, respectively. $[CH_{4(a)}]$ and $[CH_{4(b)}]$ represent the $CH_4$ mole
fractions of the sample and the background air, respectively. The intercept ($\delta_{(s)}$) of the
fit line is the isotope value of the $CH_4$ source present in the mixed sample. In linear
regression, $1/[CH_{4(a)}]$ and $\delta_{(a)}$ represent independent (X-axis) and dependent (Y-axis)
variables, respectively. This method is suitable for carbon dioxide, methane(Thom et
al., 1993), water vapor(Moreira et al., 1997), and other gases, but each gas has its



specific considerations (Pataki et al., 2003). The gas samples from each station were
collected within 30 minutes, during which the atmospheric background values (isotope
and mole fraction of $CH_4$) did not change, fulfilling the application conditions of this
method(Lu et al., 2021).

**2.5 HYSPLIT model**
The Hybrid Single-Particle Lagrange Integrated Trajectory (HYSPLIT) model
developed by the National Oceanic and Atmospheric Administration (NOAA) Air
Resources Laboratory, is a widely used public platform for different atmospheric
scales and supports online modules(Pereira et al., 2019; National Oceanic &
Atmospheric Administration, 2024b). The model has been used to calculate the air mass
transfer trajectories at different altitudes(Shan et al., 2009; Mcgowan and Clark, 2008;
Stein et al., 2015). Examples of applications include meteorological analysis of ozone
events(Shan et al., 2009), dust transport pathways(Mcgowan and Clark, 2008), dust
storm simulation(Broomandi et al., 2017; Ashrafi et al., 2014), prediction of size
distribution and mixing ratios of heavy metals in atmospheric aerosols(Chen et al., 2013)
and so on. To analyze the influence of meteorological conditions on $CH_4$ mixing ratios
and isotopes above the field station, wind direction and speed at different heights are
required, which are available from the HYSPLIT model. The time resolution of the
model could reach 1 hour and the height resolution was 1 meter. Backward trajectories
were used in this study, to calculate 24-hour backward trajectories at ground, 50 m,100
m, 200 m, and 300 m heights over each site, respectively. The input data included the
longitude and latitude of the site from field measurements and sampling time, while the
output information were wind direction and speed at different heights.

**2.6 Source partitioning with end-member mixing method**
End-member mixing method is a common method for identifying and quantifying
major sources of runoff. Several solutes have been used as tracers (typically 2-6) to
determine the contribution of each water source to total runoff. The method is based on
the mass balance of water and tracer, and the following assumptions: (1) the water



solute is constant, (2) the tracer is conservative, (3) and the source solution has an
extreme concentration(Bugaets et al., 2023; Barthold et al., 2011). Here, we applied it
to gases, using $CH_4$ mixing ratios and isotopes as tracers to investigate the contribution
of atmospheric background, open surface area, and facility area to high-altitude $CH_4$.

**2.7 Statistics**
Data analysis and graphing were performed using Origin 2024 software for Windows.
Linear fitting was based on the principle of the Least square method, indicating the 0.95
confidence intervals. A value of $P < 0.05$ was considered significant for statistical
analysis, and the fitting results are expressed as fitting mean and standard deviation.
Maximum, minimum, mean, median, outliers, and 25% -75% range values were also
analyzed and reported in the figures or tables.



## 3 Results

### 3.1 Measurements of CH₄ mixing ratios and isotopes

The CH$_4$ mixing ratios and $\delta^{13}$C-CH$_4$ values from the 11 stations in this study area ranged from 1.88 to 3.66 ppm and from -48.45‰ to -30.97‰, respectively. The maximum and minimum values of CH$_4$ isotopes were obtained at sites S2 and S6 (H: 200 m), respectively. The variation of the CH$_4$ mixing ratio and isotopic values at stations S2, S4, and S7 is significantly greater than that observed at other stations (Fig. 2). The CH$_4$ isotope and mixing ratio of the urban samples were -46.45 ± 0.49‰ and 2.04 ± 0.07 ppm, respectively. The result of direct emissions from the production well was -15.4 ± 5.72‰ and 118.98 ± 0.52 ppm, respectively (SI, Table S2). The range of ground CH$_4$ isotopic values observed at the field stations in the study area was -47.98‰ to -15.40‰, with an average of -42.96 ± 6.87‰. The CH$_4$ mixing ratio and isotopic values in the production equipment areas of the majority of sites were higher than those in the open areas. However, there were exceptions, with some field stations displaying similar values (for example, S6 and S9). The range of air CH$_4$ isotopic values of the sites in the study area was -48.45 to -44.28‰, with an average value of -46.43 ±1.08‰. The ground measurements showed higher CH$_4$ mixing ratios and isotopic values than the air, which could be an indication of CH$_4$ emissions from ONG sites (Fig. 3).

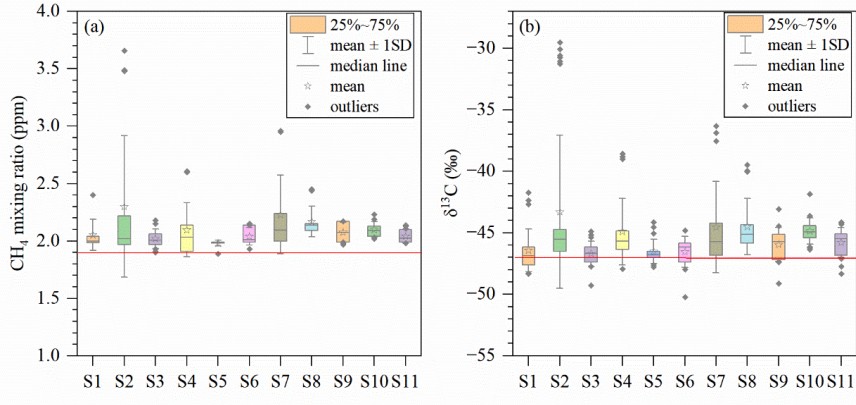

**Fig. 2** Box whisker plots of CH$_4$ mixing ratios (a) and isotopic values (b) from the studied sites (mean, median, outliers, 25% -75% range, and 1 SD are indicated in the figures; the red lines refer to CH$_4$ mixing ratios (a, 1.9 ppm) and isotopic values (b, -47‰) from the atmospheric background.





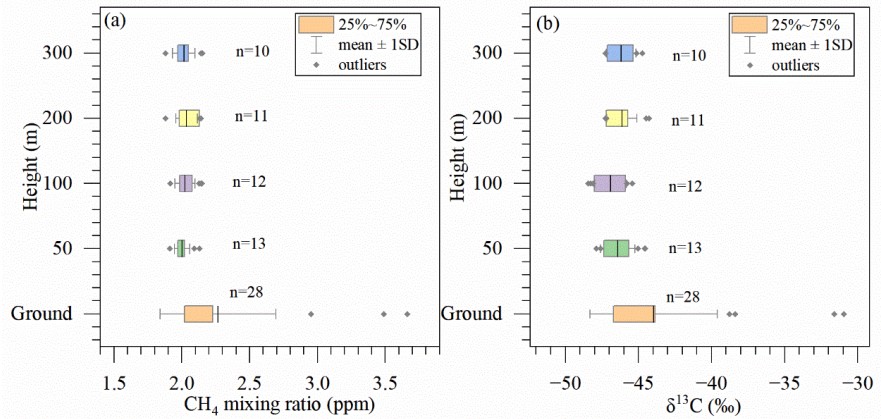

**Fig. 3** Box whisker plots showing the variations of CH$_4$ mixing ratios (a) and isotopic values (b) at different heights (from ground to 300 m at all sites); include mean, 25% - 75% range, and 1 SD; "n" represents the number of samples.

## 3.2 Vertical profiles of CH$_4$ mixing ratios and isotopes and source partitioning

The distributional trend of the CH$_4$ mixing ratios and isotopes in the vertical direction differed. For instance, the mean CH$_4$ mixing ratios were higher at 100 m than at 50 m, yet the isotopic values ($\delta^{13}$C) were lower (Fig. 3). From the perspective of a single station, the conditions were similar yet more complex. Some stations exhibited consistent trends (S1, S3, S6, S7, S9, S11), while others displayed different trends (S2, S4, S5, S8, S10) (SI, Fig. S1). For instance, the CH$_4$ mixing ratio and isotopic values at 100 m and 200 m altitude of station S8 were inversely proportional. As the altitude increased from the ground to 300 m, the CH$_4$ isotopic values of stations S4 and S1 exhibited a decline, ranging from -45.10‰ to -47.27‰ (ground to 300 m) and from -42.27‰ to -47.92‰ (ground to 100 m), respectively (SI, Fig. S1). The CH$_4$ isotopic values of stations S3 and S6 initially decreased with increasing altitude and subsequently increased, reaching a minimum at 100 m altitude (-48.45‰ and -48.11‰, respectively). The variation of the CH$_4$ isotope vertical profile at station S8 was analogous to that observed at sites S6 and S3, with the exception that the CH$_4$ isotopic minimum value reached -46.18‰ at 200 m altitude. The variation of CH$_4$ isotopic values with altitude at station S9 was complex, exhibiting a decrease followed by an increase, which then decreased again, reaching minimum and maximum values at 50 m





326 (-45.06‰) and 200 m (-44.47‰), respectively.

327  The end-member mixing method is a commonly employed technique for calculating

328 isotope mixing by various sources of GHGs(Bugaets et al., 2023). In this study, we

329 determined the contribution of $CH_4$ from the atmospheric background, surface, and

330 facility areas to the air over the sites (the details and results are presented in SI part 1).

331 The results demonstrated that the atmospheric background contributed $52.2 \pm 28.9\%$ to

332 the total, the facility area contributed $21.9 \pm 27.1\%$, and the ground open area

333 contributed $25.9 \pm 25.5\%$ (Fig. 4b). The contribution of the atmospheric background

334 initially decreased and then increased, reaching a minimum at 200 m, while the

335 contribution of the ground and facility areas initially increased and then decreased,

336 reaching a maximum at 100 m and 200 m, respectively (Fig. 4a). In cases where

337 calculations are not feasible, the $CH_4$ emissions may be predominantly influenced by

338 microbial activity or other local sources in the vicinity of the sites.

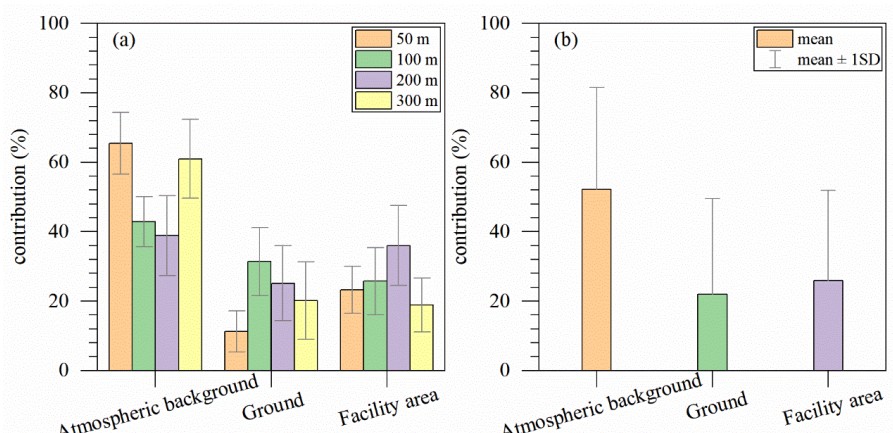

340 **Fig. 4** The fractional contributions from ambient background, surface, and facility areas
341 contribute to the vertical atmospheric sampling. (a) the proportion of contributions to
342 different heights with standard error; (b) the proportion of contributions to all heights
343 of all stations with 1SD.

345 **3.3 Characteristics of source isotopes**

346  The Keeling plot method was employed to determine the isotopic signatures ($\delta^{13}C$)

347 of $CH_4$ sources at each station, with the results presented in Fig. 5. The range of the

348 $CH_4$ source isotopic signatures varied from $-50.7 \pm 8.7‰$ to $-10.9 \pm 5.5‰$, indicating



that they were mainly thermogenic sources (associated with oil production)(Menoud et
al., 2022; Sherwood Lollar et al., 2002). The value of the $CH_4$ source signature for
station S10 was -50.7‰ ± 8.7‰, which was lower than the atmospheric background
value. Additionally, the data fitting for this field station was poor ($R^2 = 0.03$). This may
be attributed to the low $CH_4$ emissions from the site and the primary contribution of
$CH_4$ from microbial sources in the surrounding area (Table 1). Furthermore, the
measurements of the $CH_4$ samples collected in urban parks and riverside revealed that
the $CH_4$ source signature value was -35.1‰ ± 6.3‰, which was considerably higher
than the atmospheric background value. It is yet to be determined whether this value
can be considered representative of the atmospheric background value of the study area.
Additional sample data are necessary to facilitate a more comprehensive analysis. On
the other hand, the direct measurements of emission from wells indicated that source
$^{13}C$ signature was -18.7 ± 6.3‰, which is close to the result of the sample test (-15.4 ±
5.72‰) (SI, Table S2). Globally, the range of $CH_4$ isotopic values from fossil fuels is -
75‰ to -25‰, with a median value of -44‰(Defratyka et al., 2021). Our results fall
outside this range and exhibit higher values.



## 4 Discussion

### 4.1 Variations of CH$_4$ isotopic values from the atmosphere and ground

The mean values of CH$_4$ isotope were higher than the atmospheric background (-47.0 ±0.3‰) at all sites(Tyler, 1986), with some sites exhibiting values close to the atmospheric background (e.g., S1, S3, and S5). However, the average values of CH$_4$ mixing ratios   were significantly higher than the atmospheric background (1.9 ppm) at all stations(Skeie et al., 2023). This indicated that CH$_4$ emissions occurred at all sites, with obvious leakage at most stations. On the other hand, the correlation between CH$_4$ mixing ratios and isotopes at the ONG sites was significant ($R^2$=0.90). Besides, the ground exhibited a stronger correlation ($R^2$=0.94) than the air ($R^2$=0.31) (SI, Fig. S2). These findings indicated that the CH$_4$ sources in the surface of the station were rather similar and derived from the ONG industry, and that CH$_4$ in the region has a similar genesis. This conclusion was further supported by the analogous CH$_4$ source isotopic signatures from most stations determined by keeling plot approach (Fig. 5). In addition, an investigation of the potential sources of CH$_4$ in the vicinity of the ONG sites revealed that the primary source of CH$_4$ at the station was from ONG, with other sources exerting a lesser impact.

However, multiple sources may be involved when looking into the relationship between mixing ratios and isotopes at each site alone (Fig. 5). The hypothesis was corroborated by isotope data. We discovered that approximately half of the ONG production stations (6 out of 11) exhibited higher ground CH$_4$ isotopic values ($\delta^{13}$C) than those observed in the air (S1, S3, S4, S7, S8, S10). The observed decline in isotope values can be attributed to the mixing of microbial sources of CH$_4$ present in the vicinity of the stations. Conversely, the remaining half of the stations (5 out of 11) displayed ground CH$_4$ isotopes that were either lower than or comparable to those observed in the air (S2, S5, S6, S9, S11). This discrepancy may be attributed to the uncertainty associated with the sources of CH$_4$ in the air, which is more sensitive to meteorological conditions. Additionally, the majority of sites exhibit elevated CH$_4$ mixing ratios and isotopic signatures in their pipeline and facility areas. Studies have indicated that infrastructure, including components such as dehydrators, valves, compressors, and



pipelines, represents a significant source of CH$_4$ emissions from the ONG system.
Infrastructure is particularly vulnerable to CH$_4$ leakage due to corrosion and
wear(Anifowose et al., 2014; Fernandez et al., 2005; Burnham et al., 2012; Anifowose
and Odubela, 2015).

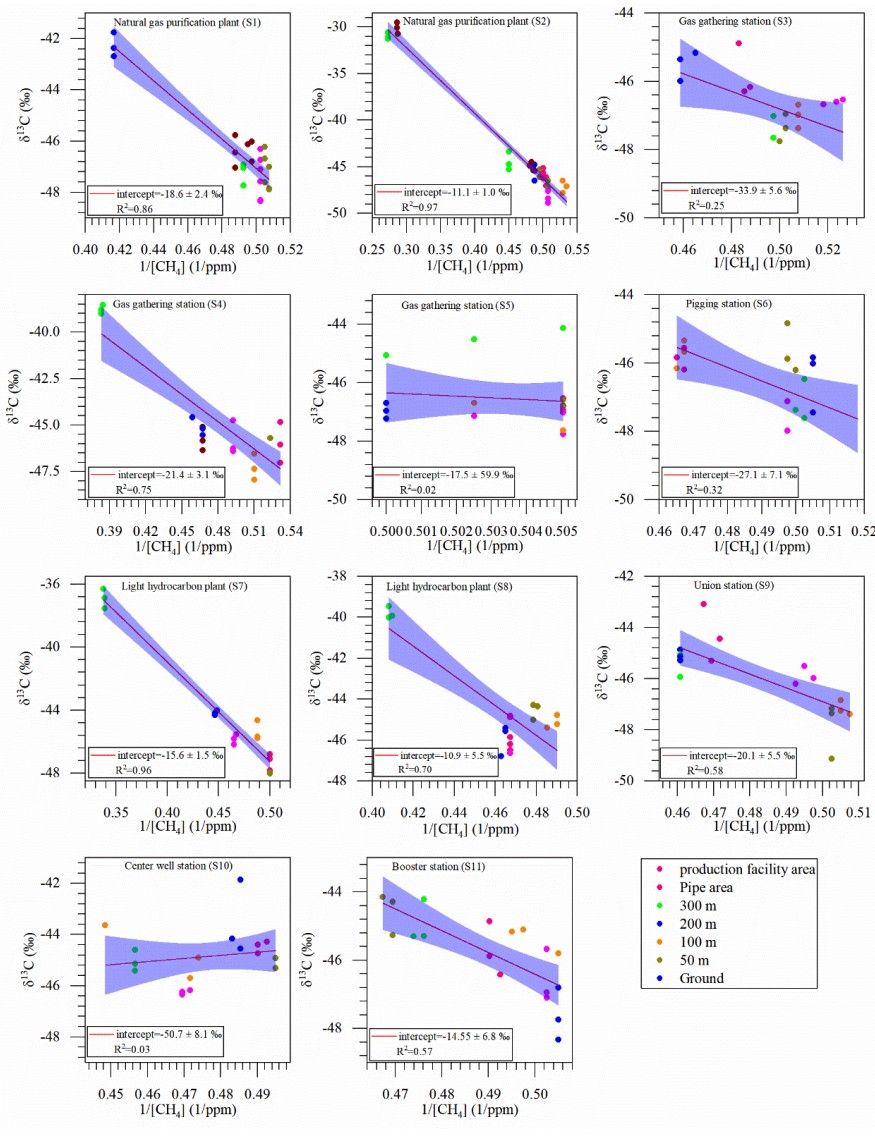


**Fig. 5** The CH$_4$ source isotopic signatures of 11 field stations. The blue area represents
the 95% confidence interval, and the red line is the result of linear regression posterior
mean fit; The samples in different positions are distinguished by different colors. The
intercept and R$^2$ are given, which means the source isotope signal value and the fitting



degree, respectively.

**4.2 Factors of drone-based isotope measurements in the atmosphere**

Significant variability was evident in the $CH_4$ mixing ratios and isotopic signatures
at the stations, with notable discrepancies observed at varying altitudes. This
inconsistency is likely attributable to a number of factors, including the presence of
additional $CH_4$ sources in the vicinity and the influence of meteorological
conditions(Kavitha and Nair, 2016). An examination of the site's environment revealed
that the area was predominantly forested, with agricultural land (paddy fields), water
bodies, and human settlements also present. Paddy fields and ponds have been
identified as the primary microbial sources of $CH_4$, characterized by lighter
isotopes(Minami and Neue, 1994; Wang et al., 2023; Vizza et al., 2022). The influence
of meteorological conditions is significant and complex, and challenging to analyze.
Wind direction and speed were obtained using the HYSPLIT model to assist with the
analysis (due to privacy considerations, the specific locations cannot be disclosed in
this work, only the data can be provided, SI, Table S3). Integration of the HYSPLIT
model data with that obtained from the meteorological station indicates that the results
produced by HYSPLIT were credible (SI, Fig. S3). The correlation analysis between
wind speed and $CH_4$ isotope results revealed an exponential relationship with an R-
squared value of 0.33 (SI, Fig. S4). This indicates that as wind speed increases, the
impact of $CH_4$ diffusion and dilution becomes more significant. Wind direction plays a
role in the uncertainty of $CH_4$ distribution, as it has a significant influence on $CH_4$
transport near the surface, resulting in a non-uniform distribution of $CH_4$ and typically
higher mixing ratios downwind from the emission source. Furthermore, upwind $CH_4$
sources can have a notable impact on $CH_4$ levels over the station. The utilization of
HYSPLIT model serves a crucial function in this regard (SI, Fig. S5 for a detailed
example of S7 site). To illustrate, the presence of a large wetland upwind can result in
air masses transporting microbial $CH_4$, consequently affecting the $CH_4$ mixing ratios
and isotopes above the station.
Moreover, the conditions at the station are among the primary determinants of the





results, encompassing factors such as the size of the sites, the treatment processes
employed, the treatment capacity, and the timing and location of sampling. A larger site
is likely to produce a great quantity of $CH_4$ emissions(Omara et al., 2016) and
accumulate $CH_4$ from a wider area, thereby exerting a more significant influence on the
air's $CH_4$ content. A correlation was observed between the area and the isotopes. It is
noteworthy that the correlation is enhanced when the site area was less than 10,000 $m^2$
(SI, Fig. S6). This may be attributed to the fact that larger site areas encompass a wider
array of factors, which can exert a more significant influence. By contrast, the treatment
processes that involve physical chemistry may exert isotope fractionation effects that
affect $CH_4$ isotopes ($\delta^{13}C$), and the general heavier isotopes of this study may be
influenced by the treatment processes. The intermittent nature of emissions from the
site facilities introduces an element of uncertainty with regard to the sampling time and
locations(Omara et al., 2016). The results of the Principal Component Analysis (PCA)
demonstrated a weak relationship among wind direction, wind speed and isotopes, and
a strong correlation between the size and capacity of the sites with $CH_4$ isotopes (SI,
Fig. S7).

**4.3 Global source isotopic signatures of ONG-derived $CH_4$**
Previous studies have investigated the characteristics of $CH_4$ isotopes in Chinese
ONG production regions, mainly in the Sichuan Basin, Xinjiang, Northeastern China,
and the Ordos Basin. Based on previous work, the reported values of $CH_4$ isotopes cover
a wide range, from -54.9% to -17.4‰ (SI, Table S1). In general, the origin of natural
gas can be divided into two major categories: biogenic and abiogenic gas(Sherwood
Lollar et al., 2002; Dai et al., 2005), of which biogenic gases include "coal-type" and
"oil-type" gases(Liu et al., 2019; Dai et al., 1985; Dai et al., 1992; Xu, 1994). Studies
have suggested that $CH_4$ of different origin carries distinct isotopic characteristics(Cai
et al., 2013; Huang et al., 2017; Wang et al., 2018; Zhang et al., 2018; Zou et al., 2007;
Zhu et al., 2014). This study is situated in the central region of the Sichuan Basin, where
previous research on $CH_4$ isotopes has predominantly concentrated on large-scale
statistical analyses. To date, no studies have specifically focused on the isotopic



464 characteristics of $CH_4$ emanating from ONG industrial sites in this area. The Sichuan

465 Basin has a complex geological environment and many gas-production layers, such as

466 Cambrian, Ordovician, Carboniferous, Jurassic, etc., and $CH_4$ from different layers with

467 various isotopic characteristics(Zhang et al., 2018; Cai et al., 2013). In comparison to

468 the findings of other researchers on $CH_4$ isotopes in the Sichuan Basin (SI, Table S1),

469 our results of $^{13}C$-$CH_4$ isotope signatures spanned more widely and also appeared to be

470 heavier. In another study, Menoud et al.(Menoud et al., 2022) examined isotopic

471 signatures of $CH_4$ from an ONG extraction plant in Romania. Their methodology aligns

472 closely with ours, and their findings indicate a range of $\delta^{13}C$ values from $-67.8 \pm 1.2$ %

473 to $-22.4 \pm 0.04$ %. Generally, our results showed heavier isotopes source signatures,

474 exceeding the global mean of fossil fuel $CH_4$ isotope (-44.0 $\pm$ 0.7‰)(Schwietzke et al.,

475 2016). This discrepancy can be attributed to a number of factors, including geographical

476 differences(Menoud et al., 2022), the treatment processing of natural gas, and the size

477 of samples. A comparison of the $CH_4$ isotopic signatures from the global ONG

478 system(Menoud et al., 2022; Lopez et al., 2017; Hoheisel et al., 2019; Defratyka et al.,

479 2021; Kang et al., 2014; Jenden et al., 1993) is presented in Fig. 6. The $\delta^{13}C$ of $CH_4$

480 was found to be lighter in the United States and Canada, but heavier in China. Regional

481 variations in $\delta^{13}C$ values were observed, even within the same region, with fluctuations

482 occurring. Our results exhibited a significantly heavier $\delta^{13}C$ than those of other studies.

483 This was attributed to differences in the origin of $CH_4$, with geographical differences

484 playing a prominent role(Zhang and Zhu, 2008; Wang et al., 2018; Defratyka et al.,

485 2021; Schoell, 1980).



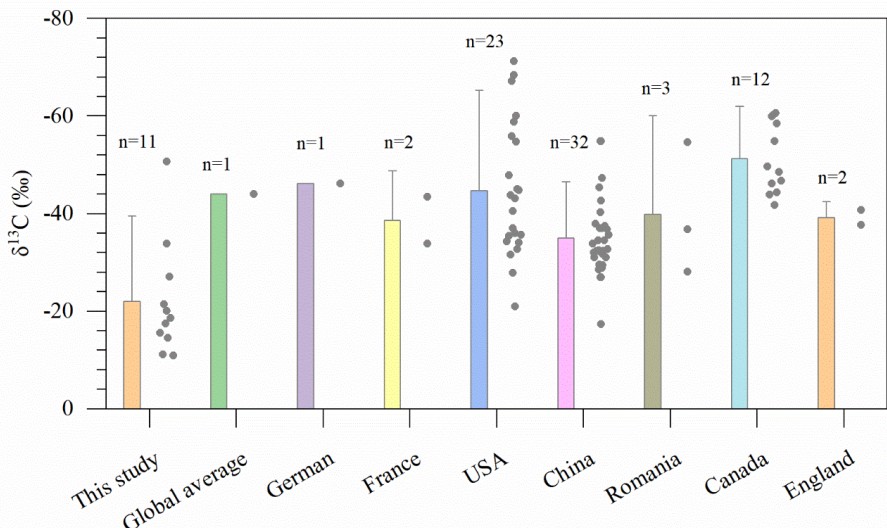

**Fig. 6** An overview of global isotopic signals of $CH_4$ emitted from ONG industry or geo-thermal sources; data from both literature and this study are included. The right side of the box chart is the data point, the number of data points is also shown at the top of the box chart, and carry on the error analysis. "n" represents the number of data points.

The $\delta^{13}C$ of $CH_4$ represents a valuable indicator for constraining and estimating $CH_4$ emissions particularly from anthropogenic sources of the globe(Milkov et al., 2020). As a sum-up, the mean $\delta^{13}C$ signatures of $CH_4$ sources as indicated from measurements of atmospheric background integrated the collective contributions from various sources of $CH_4$ (SI, Fig. S8). Hence, with updated isotopic signatures for specific sources such as ONG industry, the previous conclusions on global contribution/flux of $CH_4$ from ONG industry may need to be revised(Schwietzke et al., 2016). In comparison with previous studies, the $\delta^{13}C$ values from ONG industry in our work (-21.95‰ based on 11 stations) are significantly higher, especially different from the global flux-weighted averaged by Schwietzke et al.(Schwietzke et al., 2016). By incorporation of flux contribution from Chinese ONG industry, isotope signatures as well as global datasets utilized in the previous work(Schwietzke et al., 2016), we conducted a sensitivity analysis, examining the effect on diverse source contributions (in flux) when updated the $\delta^{13}C$-$CH_4$ from Chinese ONG industry (SI, part 2 for details). Our finding suggests



that, our field observation of isotope signature from China would elevate global fossil
fuel-derived $CH_4$ isotopes signature by about 0.5‰; as a consequence, the new result
would lead to a smaller contribution from global ONG industry (corresponds to an
overestimation of emissions by 3.47 Tg $CH_4$ $yr^{-1}$) but a larger contribution from
microbial sources. This findings is consistent with some recent research findings, such
as Chandra et al.(Chandra et al., 2024), who reported that $CH_4$ emissions decreased in
fossil fuel sources, while increasing in microbial sources during 1990-2020. In
Australia, $CH_4$ emissions from agricultural ponds (microbial sources) were
underestimated in national greenhouse gas inventories(Malerba et al., 2022). Overall,
the decline of global mean CH4 isotopic signals seem to slightly speed up in recent
years, likely supporting the importance of microbial emissions. Previous studies have
identified potential avenues for reducing $CH_4$ leakage in the ONG industry, including
improvements in technology, equipment, and management practices(Us Environmental
Protection Agency, 2012; China National Petroleum Corporation, 2023), this may
provide an insight into the overestimation of $CH_4$ emissions from ONG sources.

**4.4 Feasibility and limitations**
Atmosphere $CH_4$ isotopic research has shown its power in distinguishing between
microbial and fossil sources of global atmospheric $CH_4$ trends(Basu et al., 2022;
Bruhwiler et al., 2017). However, due to scarcity of observational evidence of various
$CH_4$ source signatures, large uncertainties still exist for such estimations. The objective
of our research was to distinguish sources of $CH_4$ as well as to quantify $CH_4$ leakage
strength at site-level, providing basic but convincing data for constraining $CH_4$ sources.
With both ground- and air-based approaches, our study has demonstrated the feasibility
of our work in studying the characteristics of $CH_4$ sources and their influencing factors
at ONG stations in SW China. Nevertheless, it is necessary to point out, that the impact
of meteorological conditions and site conditions on the dampening/masking of $CH_4$
isotope signatures in the atmosphere may be significant. Therefore, the reconciliation
between ground and atmospheric measurements as well as source partitioning remain
to be further validated, given more sampling coverage both spatially and temporally. In



addition, more sampling at different locations or different ONG plants will be greatly
beneficial to the constraints on CH$_4$ source isotope signature from fossil fuel industry
in China.



## 5 Summary


The objective of this study is to differentiate $CH_4$ sources and examine the $\delta^{13}C$
isotopic characteristics at locations where gas samples were collected from ONG
stations in the central Sichuan Basin, China. The characteristics of $CH_4$ isotopes were
analyzed on the ground, in the air, and with regard to the vertical variations of $CH_4$
isotopes. Coupled with an analysis of the surrounding environment of the stations, we
reached the conclusion that the primary source of $CH_4$ at the ONG stations is emissions
from production facilities. Furthermore, $CH_4$ on the ground at the majority of sites is
more significantly influenced by the ONG source, while CH4 in the air is more affected
by meteorological conditions. Additionally, the vertical variation of CH4 isotopes is
complex and changeable, and is affected by several factors, including meteorological
conditions, station size, sample size, and other factors. The isotopic values of $CH_4$ from
various sites were also analyzed to determine the sources of the gas. The results showed
that the isotopic values of $CH_4$ from the ONG ranged from -50.7 ± 8.7‰ to -10.9 ±
5.5‰, indicating a heavy $\delta^{13}C$ of fossil fuel. In comparison with the $CH_4$ source isotopic
values from the ONG globally, the results of this study revealed heavier isotopic
signatures. This study contributes to the global $CH_4$ isotope database from the ONG,
and addresses a gap in $CH_4$ isotope research at the site level in China. The primary
factors influencing the $CH_4$ isotope at field stations are the station's intrinsic
characteristics and the meteorological conditions present, as evidenced by the PCA
analysis. A weighted calculation on a global scale based on the results of this study
suggests that $CH_4$ emissions from microbial sources may be underestimated, while
those from fossil fuel sources may be overestimated. It is our contention that an
investigation into the isotopic characteristics of $CH_4$ at the site will prove invaluable in
distinguishing between the various sources of $CH_4$ and accounting for emissions.



## Acknowledgements


This work was financially supported by the Scientific Research Start-up Funds
(QD2022010C) from Tsinghua Shenzhen International Graduate School and Cross-
disciplinary Research and Innovation Fund Research Plan (Grant No. JC2022010) from
Tsinghua Shenzhen International Graduate School. We are grateful to the kind support
from staff members in ONG plants from SW China.

## Author contributions


D. Chen and Y. Liu conceptualized the study, with guidance from L. Yu. D. Chen and
Y. Liu conducted the field observation and sample analysis, with the help from Ze. Niu,
A, Wang and X. Pang. Y. Huang contributed data for the global literature analysis. D.
Chen, Y. Liu and L. Yu wrote the manuscript together. All coauthors reviewed and
contributed to the revision and finalization of the manuscript.

## Competing interests


The authors declare no competing interests.

## Data availability


All the data published in this work could be accessed upon request from the
corresponding author.



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
