# Peer review of "Isotopic signatures of methane emission from oil"

_EGUsphere, 2025_

## Author Comment (AC1)

**Author's response for Ref: egusphere-2025-377**

**Ref. No.:** egusphere-2025-377
**Title:** Isotopic signatures of methane emission from oil and natural gas plants in southwestern China.

**Journal**: Atmospheric Chemistry and Physics.
**Authors:** Dingxi Chen, Yi Liu, Zetong Niu, Ao Wang, Pius Otwil, Yuanyuan Huang, Zhongcong Sun, Xiaobing Pang, Liyang Zhan, and Longfei Yu.

Dear Editor,

   We would like to express our sincere appreciation to you, and the anonymous reviewers for your constructive comments and suggestions on our manuscript (egusphere-2025-377) submitted to Atmospheric Chemistry and Physics. We have carefully considered all reviewers' comments and have responded to each point in detail.

Regards,
Longfei Yu
On behalf of all coauthors
May 14, 2025

- **Reviewer comments**
- Author's response

**RC3**

1. **This manuscript presents new methane isotopic data for Chinese oil and gas infrastructure measured on a CRDS instrument. It concludes that it can distinguish between sources. Unfortunately, the data presented can only imply what the averaged fossil fuel signature might be. Overall the data shown are not convincing because: a) the emissions categories are not characterized at source, or close to individual emission points, and b) the source signature calculations mostly have very large errors and rely on a 1-point calibration of raw data.**

R: Thank you very much for your comments and critical points. We agree partially with the reviewer 3 that uncertainty in ground-based or UAV-based isotopic measurements of $CH_4$ in environmental atmosphere is generally larger than direct measurements of source isotope signatures next to singular ONG sources. However, such study design was planned on purpose, to evaluate the $CH_4$ leakage from the site-levels, which are believed to be valid for those intensive ONG processing facilities with distributed leakage points which is impossible to quantify one by one. Such work has also been conducted in other countries or regions (Menoud et al., 2022; Leitner et al., 2023; Fosco et al., 2025). In addition, we have also measured CH4 emission (with portable TDLAS instrument for CH4 mixing ratios) next to significant leakage points(Chen et al., 2024), via mobile monitoring platforms (vehicle-based monitoring on ground and UAV-based monitoring in the atmosphere). This work would provide important support to validate our isotope measurements for appointing ONG sources. In addition, for ONG sites under operation, it is highly difficult and challenging to conduct in situ monitoring or sample collections, not to mention even more restrictions to get access directly to the emission source. Therefore, our research campaign and collected data was highly valuable and scarce considering all the obstacles and restrictions in conducting systematic research from ONG entrepreneurs. Regarding close-to-source measurements, as noted in our SI (Table S2), those were the only possibilities that we could sample in body distance to the production well. And the isotope signature from this well was generally in very good consistency with our source partitioning results based on isotope measurements.

Regarding the measurements, we are sorry for the misunderstanding in our dataset and analyses. The CRDS system from Picarro used in our research has been widely applied for measuring $CH_4$ isotopic signatures from various sources (Menoud et al., 2022; Ars et al., 2024; Al-Shalan et al., 2022; Geum et al., 2024; Rella et al., 2015; Lu et al., 2021). As for the laboratory measuring our samples, the initial calibration was conducted only with 1-point calibration, mainly due to that the G2132-i system is rather stable throughout the time. Actually, we should note that, we have used two international primary standards (based on VPDB; Std1 and Std2, -68.6‰ and -40.0‰) and one

secondary standards (House-standards, Hstd, -46.89‰) which had been cross-calibrated. Along each measurement sequence (see more details as demonstrated in the figure 1), we measure all three standards together with the standards (5 samples for each sequence), during which Std 1 is used for calibration and correction for sequence-drift, Std 2 used for quality control and Hstd used for constraining long-term drift. As for our case, the measurements were finished within less than a week, so we didn't correct for long-term drift. For simplicity consideration, we only adopted Std 1 for calibration (both delta calibration and short-term drift correction). Of course, the short-term drift was minor compared with delta calibration. Nevertheless, we value the comments from reviewer 3, and will recalibration and evaluate our datasets with 2-point calibration scheme. As shown in the figure 2, given the good correlation among all three standards, we believe that our Picarro G2132-i system is rather robust and will not significantly alter our final results and main findings.

[Figure]

Fig.1 Methane Isotopic Analysis Procedure for Gas Samples

[Figure]

Fig.2 Calibration of the instrument was conducted by linear regression between the certified values and measured isotopic values of Std1 and Std2, ensuring accuracy and consistency in $\delta^{13}C$ measurements.

2.  **To emphasize further comment a) above, it is important that you give a clear distinction between the facilities that are related to oil extraction with residual gas, as this gas component could be highly fractionated, or flared on site, with those that solely focus on gas upgrading and delivery to the network, where there is likely to be less isotopic variability. Your table of facilities does not make this clear. Combustion of fossil fuels will increase $^{13}C$ in any emitted residual $CH_4$, so you need to be sure that this is not part of the activities.**

R: Thank you for your further comments. As shown in Table 1, sites S1-S2 are for Natural gas processing, including membrane separation, adsorption, desulfurization, dehydration and other processes; sites S3-S6, and S10 are for Gas Collection and transportation; sites S7-S8 are for C 3+ component of natural gas was recovered by low temperature separation process. We will do our best to add information for description of our sampling points and major facilities in the nearby. However, as indicated in the 1st response from the major comments, we are restricted from directly approaching singular sources except for two production wells (Table S2). Another source we did not measure is $CH_4$ emission from flare exhaust, which is not easily accessible for UAV or man. Hence, we do not consider the contribution from flare exhaust, but focus more on average emission scenarios from these sites. The work is designed to measure mixing atmospheric samples from the ONG site, for more effectively evaluating $CH_4$ leakage from site-level and quantifying source signatures. For more details, please refer to the 1st response to the major comments from RC3.

3. **To emphasize further comment b) above, the manuscript shows very limited understanding of the measurements and instrumentation. Calibration gases are measured by either metrology or by isotope ratio mass spectrometry and assigned a ratio. Each mass spectrometer behaves differently and so needs at least a 3-point calibration to slope correct the measured data across a range of isotopic signatures, say -75 to -20‰. The slopes for CRDS instruments show even greater calibration slope correction factors, so correcting data with one standard at -69‰ will not give the correct calibration at -10‰. Additionally, the authors refer to instrument accuracies, when they are actually showing precisions. Accuracy can only be assessed after calibration using at least 3 reference points, and applying the calibration equation to measurements of the calibration gases as unknowns. As the isotopic calibration will not change within error of measurement in the long-term it should still be possible to retrospectively provide a correction to the data by adding calibrants that are close to atmospheric background isotopic signature and with a more $^{13}$C-enriched signature.**

R: As mentioned in the 1st response to the major comments from RC3, we have indicated in full details regarding our calibration scheme and evaluations of our isotope results. We actually have laboratory standards spanning the $^{13}$C range of -68.6 ~ -40.0‰, which may not be perfect but suitable for delta calibrations during our analysis. In addition, we only have two samples from the production wells to be in the range of -15‰~ -24‰, which were not even used for keeling plot evaluations as they were next to the singular sources. Regarding the reviewers' comments on the failure of calibration for -10‰, our analytical results were indeed rarely lying in this range (95% of the data points were between -68.6 and -40.0‰). Instead, the keeling plots results were higher, with some source signatures close to the range of -20.0‰. But this won't necessarily interfere with our calibration for the isotope data of the collection air samples.

We are also aware and have the possibility of conducting 2-point calibrations for reevaluating our analytical results. As indicated previously, we will recalibrate the

isotope datasets and be more careful in interpreting our results. Regarding the mixing of precision and accuracy, we will double-check our texts and describe our analytical quality in clearer format.

**4. The authors also discuss results of vertical profiles without showing the measurement precisions, but from the calibration gas precision it is clear that for some vertical profiles the whole of the variation seen is within the measurement precision of the instrument, so these results have no meaning. This is also true for more than half of the source signature graphs presented.**

R: We thank the authors for the critical suggestions on the vertical profiles. It is important to note, for environmental atmospheric measurements, one would not expect a significant change or fluctuations of isotope signatures in $CH_4$, unless the sampling is not conducted in an open environment. Alternatively, the isotopic signatures will be clearer if next to source, but this is technically difficult and not feasible in most ONG-sites. Based on our dataset, we are able to confirm the clear and consistent source signatures of ground-based ONG $CH_4$ leakage. Certainly, there are uncertainties in the measurement itself as well as some influence of the surrounding environment or meteorological conditions. Nevertheless, based on our further detailed explanation of our analytical precision and reliability as well as the already discussed influence from meteorological conditions, we believe our results are able to reflect the $CH_4$ leakage from the ONG region in SW China.

**5. UAV sampling of air for laboratory isotopic measurement is not new, and there are papers at least back to 2016. For heights of a few hundred metres there is also the possibility to use AirCore sampling and get the full vertical profile rather than just at 3 or 4 heights.**

R: We thank the authors for the critical suggestions on the vertical profiles. It is important to note that this is the 1st study of site-level $CH_4$ leakage from Chinese ONG sites based on stable isotope techniques to our knowledge. This information could be of high value for unraveling the global debating on the variations of $CH_4$ isotope signatures from global background atmosphere. We are aware of the AirCore sampling

methods (Karion et al., 2010), which will be able to give the full vertical profile of $CH_4$

measurements. However, for the ONG sites under operation of CNPC (China National Petroleum Corporation), we only get short time windows to conduct in situ measurements, and the local site condition/facilities did not allow us to install such devices and long gas tubes for continuous measurement of $CH_4$, particularly considering these many sites we have visited (11 sites). Therefore, our study design was carefully planned based on the local situation, and the UAV method is most convenient and mobile (from site to site).

**6. There is no clear statement on what the Ground Open Area category is, or how it can be assigned an isotopic signature with any confidence. I am presuming that it is a mix of no sources, farmland sources and wetland sources, but this**

**could give a signature anywhere between -70 and -55 ‰ depending on what the sources are and their relative proportions.**

R: Thank you for your detailed comments. We are sorry for the unclear information for the surrounding environment in these rural areas. Around all ONG sites, the major land-use type is rural roads and paddy fields (scattered), through which small ditches or streams run. In those paddy fields and other rural areas, no livestock farm or landfill was present, thus excluding the possibilities of cows/pigs or waste in affecting $CH_4$ sources. In addition, during our sampling period (April), we didn't spot any biomass burning, which is also forbidden by law in China. Another important information to note is, our UAV sampling points are mostly located near middle or at least not close the edges of the ONG sites, which are much larger in area than the scattered paddy fields in the surroundings. For the revision, we will add the corresponding information to the methods and also the Table 1.

In addition, in a parallel study of our team conducted in the same region, we collected several ambient atmospheric samples for $^{13}C$-$CH_4$ close to the paddy field (1.5 m above surface, 10-20 m from the borders of paddy rice fields). The isotopic values ($\delta^{13}C$-$CH_4$= -47.2 ± 0.2‰; unpublished) were quite similar to the global background. The little influence from paddy rice on the atmospheric measurements could be due to small $CH_4$ emission from the sampling periods (relatively dry in April for Sichuan region). Secondly, all stations are surrounded by high walls, which further reduces the impact of other methane sources on near-surface methane mixing ratios.

We agree that ground open is may not be precise enough to indicate the sampling spots. A few criteria for the ground-open area: no significant observation of $CH_4$ concentration elevation compared with ambient background (Chen et al., 2024)(based on portable TDLAS methane mixing ratios monitor); at least 20 m from any facility; no spot of pipelines or valves.

7. **I also found that the selection of literature being reviewed was quite limited and focused on a few geographical areas, missing a lot of key isotopic studies on methane.**

R: Thank you for your constructive comments. We will enrich the isotopic study in the revised manuscript to cover a broader geographical area.

**Detailed Comments:**

**Abstract:**

1. **Line 33 – should have some explanation how a different isotopic signature can suggest a global overestimation.**

R: Thanks. For global estimation of $CH_4$ emission from anthropogenic activities, current methods are depending on either top-down or bottom-up estimates based on atmospheric monitoring or emission inventories. Large uncertainties remain, particularly on the debate between biological sources (wetland and so on) and industrial sources (e.g., ONG and coal). While flux measurements cannot resolve such problem

on a global basis, more and more work has recently adopted isotopic measurements to further indicate the likely contributions from either biological and industrial source(Schwietzke et al., 2016; Milkov et al., 2020). On top of such information derived from isotopes, researchers may further reduce the uncertainties in global estimation of $CH_4$ emissions. We will further elaborate this in the revision.

**Introduction:**

2. **Lines 102-104 – Start a new paragraph for the UAV topic. UAV have been used to collect samples for isotopic analysis using air sampling bags much earlier than the studies that you mention.**

R: Thanks. We agree with your comment. We will start a new paragraph for the UAV section. It is worth noting that UAV-based sampling is an earlier-developed technique; however, our intention is to emphasize the application of UAV monitoring at the station scale.

In the revision, we will re-screen the literature to cover the classic references.

3. **Lines 104-106 – Don't mix up UAV and large aircraft sampling. France et al. 2021 used a large aircraft, not a UAV.**

R: Thanks. We agree with your comment, and we will make the revision.

4. **Line 108 – studies concentrated in foreign countries. This statement depends on who is reading the text. You mean in countries outside of China.**

R: Thanks. We sincerely apologize for the lack of clarity in our wording; we will make the necessary revisions in the updated manuscript.

5. **Lines 111-114 – an isotopic signature is that which is calculated for the source. It cannot be used for the measurement of an atmospheric mixture. The signature will not change due to meteorology or sampling method. The measured values can be different due to sampling distance from the source and dispersion meaning that there is a larger component of the ambient air in the mixture, but the source signature will remain the same.**

R: Thanks. We agree with your comment. We sincerely apologize for the confusion in the description of isotope data. We agree that isotope signature reflects the source itself, but not the mixing isotope results. We will revise the manuscript accordingly.

6. **Lines 125-126 – what do you mean by Upper Atmosphere? You sampled at 200-300m. This is not upper atmosphere. That would be the mesosphere.**

R: Thanks. In this study, the term "upper atmosphere" refers to the 50 m, 100 m, 200 m, and 300 m heights. Alternatively, this refers to the atmosphere above these O&G plants. We will correct this in the revision.

**Method:**

**7. Table 1 – needs better formatting to avoid splitting words between lines. The column on surrounding environment needs more detail for farmland. Does it contain rice growing, ruminants or both?**

R: Thanks for the valuable comments. We will improve our formatting regarding the tables. For surrounding environment, as we noted previously in the response, it was mainly rural roads and paddy fields (scattered), through which small ditches or streams run. In those paddy fields and other rural areas, no livestock farm or landfill was present, thus excluding the possibilities of cows/pigs or waste in affecting $CH_4$ sources. For more details, please refer to our response to RC1.

**8. Figure 1 – needs a better picture to show how the bags and pump are connected to each other and to the UAV and to the sampling inlet. Perhaps drawing a schematic would be better than the top right picture.**

R: Thanks for the valuable comments. We will draw a schematic to show the UAV sampling system.

**9. Line 207 – as above, 200-300m is not high altitude.**

R: Thanks for your comments. We will provide a more accurate description.

**10. Lines 213-215 – precision not accuracy; you can only get accuracy if you calibrate against isotopic standards of known composition across the known measurement range. The Picarro G2132 is a discontinued instrument before 2024. Are these precisions quoted from Picarro for the G2132 or the newer replacement G2201i?**

R: Thanks for your comments. We will change "accuracy" to "precision". These precisions were quoted from the Picarro G2132 model, originating from the manufacturer.

**11. Line 216 - you cannot have a 1-point calibration for isotopes, as the isotopic calibration slope is very different for the CRDS between say -70 and -30‰, probably making a difference of about 1‰ for every 10‰ change in measured isotopic value compared to the slope defined by metrology and isotope ratio mass spectrometry. Therefore, if you calibrate at -70‰ you will be then incorrect by 4‰ at -30‰. You need at least 3 calibration points across your range of measured values.**

R: As mentioned in the 1st response to the major comments from RC3, we have indicated in full details regarding our calibration scheme and evaluations of our isotope results. We actually have laboratory standards spanning the $^{13}$C range of -68.6 ~ -40.0‰, which may not be perfect but suitable for delta calibrations during our analysis. In addition, we only have two samples from the production wells to be in the range of -15‰~ -24‰, which were not even used for keeling plot evaluations as they were next to the singular sources. Regarding the reviewers' comments on the failure of calibration

for -10‰, our analytical results were indeed rarely lying in this range (95% of the data points were between -68.6 and -40.0‰). Instead, the keeling plots results were higher, with some source signatures close to the range of -20.0‰. But this won't necessarily interfere with our calibration for the isotope data of the collection air samples.

We are also aware and have the possibility of conducting 2-point calibrations for reevaluating our analytical results. As indicated previously, we will recalibrate the isotope datasets and be more careful in interpreting our results. Regarding the mixing of precision and accuracy, we will double-check our texts and describe our analytical quality in clearer format.

**12. Lines 257-258 – need to check some of the writing to avoid sentences like 'Backward trajectories were used ….. to calculate ….. back trajectories'.**

R: Thanks for your comments. We will review and revise accordingly.

**Results:**

**13. Line 283 - this is not the range shown on box and whisker plots (1.7 ppm and below -50‰).**

R: Thanks for your comments. We will revise the box and whisker plots.

**14. Figure 2 - how do you measure a $CH_4$ mixing ratio of 1.7 ppm that is well below the lowest measurement at any background site, using South Pole as the lowest?**

R: Thanks for your comments. The lower end of the box-whisker plot was representing the large variation range of the sampling points, not necessarily indicating that the measured mixing ratio was 1.7 ppm. Instead, Table S2 shows all the results, with a minimum value of $CH_4$ mixing ratio to be near 1.88 ppm.

**15. Lines 302-321 - you cannot make statements about changes with altitude when they are smaller than the instrument measurement precision over the measurement period of 120 sec, which at near background mixing ratios (1.9-2.2 ppm) is about ±1‰ at best. Also, you should show the precisions on these values. Your isotopic data will be compared to those made by IRMS at <0.1‰, so the reader needs to believe that your measurements are actually showing real variations and not just measurement noise.**

R: As mentioned in the 1st response to the major comments from RC3, we believe that our Picarro G2132-i system is rather reliable and would allow us to interpret the variations along altitude. We would show the precision in the revision.

**16. Lines 332-333 – you do not explain what is ground open area (also no explanation in the supplementary material), so what is it and how can you realistically assign an isotopic signature to it? Are we to presume that this is a mixture of no sources, natural wetland sources and farmland sources?**

R: Please refer to the 6th response to the major comments from RC3 for details.

**Discussion**

**17. Figure 5 – a problem with the low precision of CRDS for isotopes at close to background mixing ratios means that you could put any slope through most of these graphs, particularly as you have avoided to show any error bars on the measurements. Only those with trend errors of <2.5‰ are strongly correlated but again mostly by joining 2 clusters rather than a series of points along the line, an important factor to reduce source signature calculation errors.**

R: We fully agree with the reviewer that, if data points are not evenly spread along the line, the estimation of source isotopic signatures with keeling plot approach may be relatively more uncertain, which is also the major drawback from the classic keel plot approach. However, in actual field applications, the interpretation is based on the natural extent of data collected by our sampling campaign and could not be manipulated for perfect regression. Therefore, as we indicated in the previous response to RC3, we believe that our data analysis and interpretation of source isotope signature is valuable and constructive for understanding O&G CH4 emission sources, as long as we prove that our analytical precision and sampling method is reasonable. In the revision, we will further incorporate the errors from measurements and provide more quality assurance/discussion on the keel plot evaluation.

**18. Lines 422-424 – the statistical analysis in Figure S4 is not valid as the interpretation hinges on 1 point and without it there would be no correlation.**

R: We agree that such regression may be uncertain given the wide spread of all data points. However, we can not simply remove this point with replicate measurements from a field measurement study, as we believe the spread of data points as found in our *in situ* work is important and valid for interpretating the pattern particularly for field practice. In the revision, we will describe such relationship based on Fig. S4 with more cautions.

**19. Lines 425-427 - presumably you measured upwind of each study area and collected a sample before sampling on site. If there was an influence from a wetland source for example you would see a higher CH4 background and a depletion in ¹³C.**

R: Thanks for your comments. We fully agree with your comment. As described later in the manuscript, we used the S7 station as an example to illustrate the impact of wind direction on methane distribution. Based on our results, the wind effect was mostly a connection with the background atmosphere and not mixing with another source (higher CH4 mixing ratio).

**20. Lines 435-437 - you should specify that this statement is for fossil fuel extraction sites. This does not necessarily apply to other sources, and a big pipeline leak (500-1000 kg/hr) can come from a small point source.**

R: Thanks for your comments. We will specify it in the revision. It is true that, for fossil fuel extraction sites or even major pipeline leaks, a small point would result in very

large emission, leaving the site size to be less important. In the revision, we will mention this together with the discussion of site size effect on the measurements.

**21. Lines 471-485 - the isotopes of small % gas residue from an oil well are easily fractionated. Purely gas production wells tend to have much more homogenised signatures that are better correlated with the temperature that the gas is produced at, the hotter the gas production window the more enriched in $^{13}$C it is.**

R: Thank you for your comment. We agree that the enrichment of methane isotopes is related to temperature. Due to different formation conditions, methane from different geographical locations can exhibit significant differences in the concentration of $^{13}$C.

**22. Figure 6 - you should say which studies you used for this compilation. There are far more studies outside of North America and China that you do not show here, so what you show is not representative. Additionally, ONG covers a lot of different source types. How can you be sure that you are comparing like with like? So, it is not an overview figure. If you are using this data you should focus on oil and gas production areas (which includes Romania), but not include studies on refined gas distribution networks in European cities.**

R: Thank you for your valuable comment. We will further enrich our coverage of literature and data sources in more regions, providing a detail table indicating the data sources (including whether the site is a production site or processing site). We need to clarify that the collected values were all from oil and gas production areas, without including studies on refined gas distribution networks.

**23. Lines 507-511 - you cannot pluck this statement out of the air with no explanation of how it is calculated (and even with the SI it is not very clear what are your defined parameters), and you should not compare gas as an oil field residual product, with gas production wells, or with a global average of -44‰ that includes coal, oil and gas sources with a range of 50‰ or more.**

R: Thank you for your comment. Firstly, we will revise the description in the SI to make it clearer and easier to understand.

We thank the reviewer again for the very constructive suggestion. We do agree that it is best to separate $CH_4$ isotopic signature for production and processing sites; however, due to limited data and site coverage in one project, this is difficult and not practical. In addition, we used our measurements to represent the $CH_4$ isotope signature from Chinese O&G industry, mainly because this is the very first site-level study and may not be comparable with other studies investigating geological sources. On the global basis, the studies on $CH_4$ isotope from coal, oil and natural gas industry are rather scare. In addition, the sensitivity practice was mainly used for indicating how $CH_4$ isotope signature evaluation from Chinese O&G industry may assist in flux estimations; more work is certainly necessary and indispensable for better constraining $CH_4$ emission estimates from Chinese O&G activities. Methane source isotopic data with broader coverage and higher resolution (including methane from all sources) will contribute to

global methane emission inversion and lead to more accurate emission estimates.

**24. Lines 514-515 - what is the relevance of this sentence?   Yes, ponds have been overlooked, but most sources are not correct in the inventories, and there is much more work needed to reduce the uncertainties.**

R: We sorry for the unclear wording that caused the misunderstanding. Our intention was to highlight that microbial methane has been underestimated in the accounting inventory, which is consistent with the findings of this study and serves to confirm the results presented in the manuscript. We will leave this out in the revision.

**25. Line 521 - who says that it is an overestimation?   If you increase biogenic sources by 30% and fossil fuel sources by 10% the result is a decrease in global $^{13}$C.**

R: Thank you for your valuable criticism. If you vary both biogenic and fossil fuel sources, the results will be different, and can even result in different directions. However, our sensitivity analysis was conducted based on the hypothesis that the global mean isotope signatures of $CH_4$ from biogenic sources remain constant. Therefore, to avoid confusion, we will clearly indicate that our sensitivity analysis is based on an ideal scenario.

**26. Lines 524-525 - it does a lot more than just separate microbial and fossil sources and can distinguish a wide range of biogenic processes, such as between oxidation and reduction, between different feedstocks and diets. Microbial from fossil is just scratching the surface.**

R: Thank you very much for your comment. We understand that more processes are interplaying in the regulation of global $CH_4$ isotopes and trends. Nevertheless, much remains unknown and is still complexed in the global $CH_4$ science community. Such topics may be out of the scope for our current work.

**27. Line 529 - Unfortunately the data shown are not convincing, as the emissions categories are not characterized at source, or close to individual emission points.**

R: Thank you for your critical comments. As mentioned in the 1st response to the major comments from RC3. We should note that we had a few samples collected near production wells, which is close to individual emission points. We believe, as we noted in the responses to RCs, our *in situ* work provides important field evidence of $CH_4$ emission and isotope characteristics from the O&G sites in China, and will likely bring significant scientific value to the research community.

**Summary:**

**28. Line 541 – Unfortunately the work does not distinguish between sources, it only implies what the averaged fossil fuel signature might be.**

R: Thank you for your feedback. We will clarify our descriptions in the summary. We

agree that we mainly examined the $CH_4$ source from O&G industry, and this is not related with multiple source partitioning. The original purpose of source distinguishing refers to the tracing of source from different production units or the comparison of sites.

**29. Line 556 - The global CH₄ isotope databases of Schweitzke, Sherwood and Menoud contain measurements from direct sampling of sources and measurement by isotope ratio mass spectrometry to high precision. Measurements of source signatures with large errors should not be included in any database that will be used for global and regional modelling.**

R: We believe that our isotope data quality is useful for indicating the $CH_4$ emission patterns as well as source characteristics from Chinese O&G sites, as mentioned for several times in the responses to RC3. Also referred to the response for line 33, we used the isotope results from our study to mainly indicate the sensitivity of $CH_4$ isotope data in estimating industry $CH_4$ emission from oil and natural gas production/processing. This is similar to other global or regional studies(Fisher et al., 2017; Nisbet et al., 2016).

**References:**

**30. Burnham et al. is not correctly formatted.**
R: Thanks. We will revise it.

**31. There are 9 references to web sites, all ending in the word 'last' and all incorrectly placed within the list.**
R: Thanks. We will revise it.

**Supplementary Data:**

**32. S2 – how robust is the signature of -21.95‰? Signature is not based on correctly calibrated data so how do you know that it is typical? The precisions on many of the 11 station source calculations are very wide so these should not be used in averaging. Global fossil fuel includes a significant component of coal, whereas your ONG sites do not, so is this a good comparison?**

R: We apologize for the confusion particularly to the reviewer. We have explained in much detail in the previous replies, that our calibration is reliable for interpretating the source isotope signatures, although we unfortunately omitted the details in the earlier version. The overall measurement precisions for samples across the 11 sites were actually consistent and follow similar source characteristics as validated in the keel plot with all ground data points (-17.8‰; see the figure 3 below). In the revision, we will use the isotope signature obtained from the overall keel plot estimates and assess uncertainties in our sensitivity analysis.

[Figure]

Fig.3 Analysis of Methane Source Isotopic Signature. The Keeling plot approach was applied using ground-level sample data from all stations.

**33. S2 – you correctly use Saunois et al., 2024 here for the budget calculations, but in the main paper you use the older Saunois et al 2016 which has the budget to 2012.**

R: Thanks. We will revise it to ensure the use of the latest data and maintain consistency.

**34. Table S1 – basins are normally predominantly gas only, or oil with a bit of gas, and the gas only tend to have narrower isotopic ranges typical of mature gas. It would be useful to know what is the proportion of oil to gas extraction in these basins.**

R: Thank you for your comment. We will make every effort to seek out relevant data.

**35. Table S2 – Time column - this is not a time and looks more like a date. Presume 13 April. Need to show the sampling height for pipeline and production areas.**

R: Thanks. We will change "time" to "date". Regarding the sampling height, it is indicated in Table S2—for example, "H:100 m" denotes a sampling height of 100 meters.

**36. Table S4 – I still have no idea what is Ground and how it can be assigned an isotopic signature.**

R: As mentioned in the 6[th] response to the major comments from RC3.

**37. Figure S4 - this statistical interpretation is not valid. Remove 1 data point and there is no correlation.**

R: Please see our response 18 to RC3 as referred to line 422-424.

**38. Figure S5 - so why was the plume to the SW of site S7 not sampled for isotopic analysis, to get a more precise source signature? The plume could have been**

**sampled at multiple points to give a spread of data on the Keeling plot and points above 10 ppm with much better CRDS precision.**

R: Thank you very much for your constructive comment. It would be great if we could sample paired isotope data together with the plume measurement. However, this was conducted by another instrument by the other research team focusing on vehicle-based mobile monitoring (the data shown was collected at static mode).

39. **Figure S8 – why is this included? The global trend is not part of the study and this trend has been reported in many other studies that have thoroughly analysed the reasons behind this trend.**

R: Thank you for your comment. Figure S8 in our Supplementary Information is intended to provide background for our analytical calculations and to help better contextualize and understand our hypothetical estimates. We will remove it and only mention this in the contents instead.

**Reference**

Al-Shalan, A., Lowry, D., Fisher, R. E., Nisbet, E. G., Zazzeri, G., Al-Sarawi, M., and France, J. L.: Methane emissions in Kuwait: Plume identification, isotopic characterisation and inventory verification, Atmospheric Environment, 268, 118763, https://doi.org/10.1016/j.atmosenv.2021.118763, 2022.

Ars, S., Arismendi, G. G., Muehlenbachs, K., Worthy, D. E. J., and Vogel, F.: Using in situ measurements of $\delta^{13}C$ in methane to investigate methane emissions from the western Canada sedimentary basin, Atmospheric Environment: X, 23, 100286, https://doi.org/10.1016/j.aeaoa.2024.100286, 2024.

Chen, L., Pang, X., Wu, Z., Huang, R., Hu, J., Liu, Y., Zhou, L., Zhou, J., and Wang, Z.: Unmanned aerial vehicles equipped with sensor packages to study spatiotemporal variations of air pollutants in industry parks, Philosophical Transactions of the Royal Society A: Mathematical, Physical and Engineering Sciences, 382, 20230314, doi:10.1098/rsta.2023.0314, 2024.

Fisher, R. E., France, J. L., Lowry, D., Lanoisellé, M., Brownlow, R., Pyle, J. A., Cain, M., Warwick, N., Skiba, U. M., Drewer, J., Dinsmore, K. J., Leeson, S. R., Bauguitte, S. J.-B., Wellpott, A., O'Shea, S. J., Allen, G., Gallagher, M. W., Pitt, J., Percival, C. J., Bower, K., George, C., Hayman, G. D., Aalto, T., Lohila, A., Aurela, M., Laurila, T., Crill, P. M., McCalley, C. K., and Nisbet, E. G.: Measurement of the 13C isotopic signature of methane emissions from northern European wetlands, Global Biogeochemical Cycles, 31, 605-623, https://doi.org/10.1002/2016GB005504, 2017.

Fosco, D., Molfetta, M. D., Renzulli, P., Notarnicola, B., Carella, C., and Fedele, G.: Innovative drone-based methodology for quantifying methane emissions from landfills, Waste Management, 195, 79-91, https://doi.org/10.1016/j.wasman.2025.01.033, 2025.

Geum, S., Park, H., Choi, H., Kim, Y., Lee, H., Joo, S., Oh, Y.-S., Michel, S. E., and Park, S.: Identifying emission sources of CH4 in East Asia based on in-situ observations of atmospheric δ13C-CH4 and C2H6, Science of The Total Environment, 908, 168433, https://doi.org/10.1016/j.scitotenv.2023.168433, 2024.

Karion, A., Sweeney, C., Tans, P., and Newberger, T.: AirCore: An Innovative Atmospheric Sampling System, Journal of Atmospheric and Oceanic Technology, 27, 1839-1853, https://doi.org/10.1175/2010JTECHA1448.1, 2010.

Leitner, S., Feichtinger, W., Mayer, S., Mayer, F., Krompetz, D., Hood-Nowotny, R., and Watzinger, A.: UAV-based sampling systems to analyse greenhouse gases and volatile organic compounds encompassing compound-specific stable isotope analysis, Atmos. Meas. Tech., 16, 513-527, 10.5194/amt-16-513-2023, 2023.

Lu, X., Harris, S. J., Fisher, R. E., France, J. L., Nisbet, E. G., Lowry, D., Röckmann, T., van der Veen, C., Menoud, M., Schwietzke, S., and Kelly, B. F. J.: Isotopic signatures of major methane sources in the coal seam gas fields and adjacent agricultural districts, Queensland, Australia, Atmos. Chem. Phys., 21, 10527-10555, 2021.

Menoud, M., van der Veen, C., Maazallahi, H., Hensen, A., Velzeboer, I., van den Bulk, P., Delre, A., Korben, P., Schwietzke, S., Ardelean, M., Calcan, A., Etiope, G., Baciu, C., Scheutz, C., Schmidt, M., and Röckmann, T.: $CH_4$ isotopic signatures of emissions from oil and gas extraction sites in Romania, Elementa: Science of the Anthropocene, 10, 2022.

Milkov, A. V., Schwietzke, S., Allen, G., Sherwood, O. A., and Etiope, G.: Using global isotopic data to constrain the role of shale gas production in recent increases in atmospheric methane, Scientific Reports, 10, 4199, 2020.

Nisbet, E. G., Dlugokencky, E. J., Manning, M. R., Lowry, D., Fisher, R. E., France, J. L., Michel, S. E., Miller, J. B., White, J. W. C., Vaughn, B., Bousquet, P., Pyle, J. A., Warwick, N. J., Cain, M., Brownlow, R., Zazzeri, G., Lanoisellé, M., Manning, A. C., Gloor, E., Worthy, D. E. J., Brunke, E.-G., Labuschagne, C., Wolff, E. W., and Ganesan, A. L.: Rising atmospheric methane: 2007–2014 growth and isotopic shift, Global Biogeochemical Cycles, 30, 1356-1370, https://doi.org/10.1002/2016GB005406, 2016.

Rella, C. W., Hoffnagle, J., He, Y., and Tajima, S.: Local- and regional-scale measurements of $CH_4$, $\delta^{13}CH_4$, and $C_2H_6$ in the Uintah Basin using a mobile stable isotope analyzer, Atmos. Meas. Tech., 8, 4539-4559, 10.5194/amt-8-4539-2015, 2015.

Schwietzke, S., Sherwood, O. A., Bruhwiler, L. M. P., Miller, J. B., Etiope, G., Dlugokencky, E. J., Michel, S. E., Arling, V. A., Vaughn, B. H., White, J. W. C., and Tans, P. P.: Upward revision of global fossil fuel methane emissions based on isotope database, Nature, 538, 88-91, 2016.

---

## Author Comment (AC2)

**Author's response for Ref: egusphere-2025-377**

**Ref. No.:** egusphere-2025-377
**Title:** Isotopic signatures of methane emission from oil and natural gas plants in southwestern China.

**Journal**: Atmospheric Chemistry and Physics.
**Authors:** Dingxi Chen, Yi Liu, Zetong Niu, Ao Wang, Pius Otwil, Yuanyuan Huang, Zhongcong Sun, Xiaobing Pang, Liyang Zhan, and Longfei Yu.

Dear Editor,

    We would like to express our sincere appreciation to you, and the anonymous reviewers for your constructive comments and suggestions on our manuscript (egusphere-2025-377) submitted to Atmospheric Chemistry and Physics. We have carefully considered all reviewers' comments and have responded to each point in detail.

Regards,
Longfei Yu
On behalf of all coauthors
May 14, 2025

- **Reviewer comments**
- Author's response

**RC1**

**General comments:**

**This paper provides important new isotopic measurements from methane emitted by China's Oil and Gas sector. China is the world's largest emitter of anthropogenic methane and isotopic data are essential if Chinese emissions are to be quantified by sector. Thus these new measurements are very valuable indeed. The paper should certainly be published. That said, there are a number of problems with the manuscript as it stands at the moment and it needs to be revised before final acceptance.**

R: Thanks for your valuable comments and suggestions. Please find our detailed response below.

**Specific points:**

**1.The introduction needs to be heavily rewritten between lines 37-107. It reads rather like something written at the start of the project some years ago and lightly updated. This is a very active field and many important recent references are missing, while a lot of good but very elderly papers are still cited. I would strongly suggest shortening this section (L37-107) by perhaps half and making it much more modern. I have added a list of papers that might be considered below. In particular I would draw attention to the ongoing work by Saunois et al, most recently in 2024/5. Maybe the International Energy Agency should be cited for China's total methane emissions. The state of methane should be updated to 2024 – see Michel et al. 2024 and Nisbet et al 2025. Given the focus on field measurement of isotopes, maybe there is one older reference (Dlugokencky et al. 2011) but many new papers.**

R: Thanks for your suggestion. Regarding the introduction section from L37 to L107, we agree that more citations should be provided. We will further revise it and improve the conciseness.

**2.From line 106-129 the introduction gets specific. That's good, but maybe there should be a paragraph on the power of isotopes.**

R: Good idea. We have actually indicated the potential of $CH_4$ isotope techniques in quantifying CH4 source contributions in L69-76, before we move on to the introduction of ONG-related $CH_4$ monitoring or assessment. In the revision, we will add contents describing the application of methane isotopes in assessing $CH_4$ emissions from ONG industry.

**3.Line 148 – Paddy fields – more information needed here. This is important because the isotopes help discriminate between ricefield methane and fossil methane. Also, how many cows and how much pig manure is in the region, and how many landfills. Another major factor is biomass burning, that can give very heavy methane (as in some later results in the paper).**

R: Thank you for your detailed comments. We are sorry for the unclear information for the surrounding environment in these rural areas. Around all ONG sites, the major land-use type is rural roads and paddy fields (scattered), through which small ditches or streams run. In those paddy fields and other rural areas, no livestock farm or landfill was present, thus excluding the possibilities of cows/pigs or waste in affecting $CH_4$ sources. In addition, during our sampling period (April), we didn't spot any biomass burning, which is also forbidden by law in China. Another important information to note is, our UAV sampling points are mostly located near middle or at least not close the edges of the ONG sites, which are much larger in area than the scattered paddy fields in the surroundings. For the revision, we will add the corresponding information to the methods and also the Table 1.

In addition, in a parallel study of our team conducted in the same region, we collected several ambient atmospheric samples for $^{13}C-CH_4$ close to the paddy field (1.5 m above surface, 10-20 m from the borders of paddy rice fields). The isotopic values ($\delta^{13}C-CH_4$= -47.2 ± 0.2‰; unpublished) were quite similar to the global background. The little influence from paddy rice on the atmospheric measurements could be due to small $CH_4$ emission from the sampling periods (relatively dry in April for Sichuan region).

In the revision, we will discuss more carefully on the possible influence of surrounding environment on our isotope measurements based on UAV sampling.

**4.Line 167 – maybe have a paragraph break here.**

R: Thank you for your comment. Paragraph separation has been made accordingly.

**5.Line 196 to 208 in Section 2.2 – no information is given about time of day and diurnal variation in the height of the boundary layer, yet this is obviously important to the later discussion.**

R: Thank you for your comment. During our sampling campaign, the air sample collections based on UAV were conducted consistently around the noon time following consecutive days, as this is also according the regulations from ONG site managements (for external visitors). Mostly, the samples were taken between 11:00 am and 2:00 pm. We will indicate the time in the methods. Regarding the diurnal variation in the height of the boundary layer, we will add the information in the methods as well as in the discussion. However, we believe there won't be too much difference considering the short time window for our sampling.

**6.Line 243-261 Hysplit - How much local diurnal understanding is there for**

**the movement of the boundary layer? Is there any information about the stability of the air masses during UAV sampling? Pasquill stability classes?**

R: Thank you for your comment. This is a very good suggestion. As we noted previously, our sampling was conducted all in the daytime, along a rather short time window. Therefore, diurnal variability of boundary layer would not likely exert a large impact on the air mixing. We have relooked into our Hysplit-model analysis, and further computed stability index. It shows that the Pasquill stability class during UAV sampling predominantly was C (Slightly unstable conditions).

**7.Line 228 – Keeling plot. What line regression is being used? Maybe see Akritas and Bershady (1996) as used in France et al 2016 (see below for details)**

R: Thank you for your comment. We will indicate more clearly in the method that "general linear regression method is used for keeling plot calculations, while the uncertainties were evaluated based on the ordinary least squares (OLS) method".

**8.Section 3.1 is the core of the paper and very valuable.**

R: Thank you very much for your comments and recognition.

**9.Section 3.2 has no mention of time of day or diurnal evolution of the boundary layer. Also there is no real discussion of other local sources including rice and animals (isotopically light) and crop waste and other biomass fires (heavy). Some of the heavy values (e.g. in L356 could be from local fires. However the very heavy value directly measured in L361 is indeed interesting. Overall I think this section 3.2 of the paper needs a fairly major reevaluation.**

R: Thank you for your comment. We fully agree that we should add more discussions on the variabilities of boundary layer regarding vertical mixing conditions as well as other sources contributed from non-ONG fields. For more details, please refer to our earlier response to RC1.

For the exceptionally heavy value as found for the well (YJ-01&02; Table S2), it was sampled next to leakage source (ground) and also observed with high CH4 mixing ratio, confirming that this is not influenced by surrounding environment. In addition, all the ONG sites were fenced with high walls for security reason.

Overall, the isotopic source signature of CH4 leakage from our studied sites is generally consistent either for individual sites or total average, supporting that the ONG sites in SW China represents a $CH_4$ leakage source that is distinctive in isotope signals. This can be attributed to both the geothermal sources or processing procedures from the local.

**10.Line 331 percentages are quoted to a precision far beyond the real uncertainty. About half and about a fifth to a quarter might be a more accurate**

**statement.**

R: Thank you very much for your comments and suggestions. The necessary revisions will be made in the manuscript to make the description more accurate. We will also be more cautious in the discussion.

**11.Line 365 onwards. The discussion should take into account other local sources – rice, animals, fires, and perhaps coal use. Fig 5 would be useful also a Table. Line 401 linear regression method not specified – see France et al / Ahritas and Bershady method.**

R: Thank you very much for your comments and suggestions.

We have discussed these points in details in the previous responses to RC1. Regarding the local sources in the surrounding environment, we have supplemented additional information and descriptions. Overall, there is likely little influence from the surrounding environment (few residences and no biomass burning spotted).

Regarding the regression methods for keeling plot evaluations, we have indicated that "general linear regression method is used for keeling plot calculations, while the uncertainties were evaluated based on the ordinary least squares (OLS) method".

**12.Line 451 onwards – global comparison – see references below.**

R: Thanks for your suggestion. We will consider the references and enrich our comparison with other relevant studies.

**CONCLUSION**

**This paper present important new results that will be very useful in attributing China's methane emissions to specific sources. The work should certainly be published. But the paper needs some work still.**

**REFERENCES to consider: don't cite all but pick and choose which fit best in the text as it is revised.**

**SPECIFIC Oil and gas and Keeling:**

**Al-Shalan, Aliah, et al. "Methane emissions in Kuwait: Plume identification, isotopic characterisation and inventory verification."** *Atmospheric Environment* **268 (2022): 118763.**

**Akritas, M. G., and M. A. Bershady (1996), Linear regression for astronomical data with measurement errors and intrinsic scatter, Astrophys. J., 470(2), 706–714, doi:10.1086/177901.**

**Andersen, Truls, et al. "Local to regional methane emissions from the Upper Silesia Coal Basin (USCB) quantified using UAV-based atmospheric measurements."** *Atmospheric Chemistry and Physics* **https://doi.org/10.5194/acp-23-5191-2023**

**Ars, Sébastien, et al. "Using in situ measurements of δ13C in methane to investigate methane emissions from the western Canada sedimentary**

basin." *Atmospheric Environment: X* 23 (2024): 100286.

Chen, Zichong, et al. "Methane emissions from China: a high-resolution inversion of TROPOMI satellite observations." *Atmospheric Chemistry and Physics* 22.16 (2022): 10809-10826.

Dlugokencky, Edward J., et al. "Global atmospheric methane: budget, changes and dangers." *Philosophical Transactions of the Royal Society A: Mathematical, Physical and Engineering Sciences* 369.1943 (2011): 2058-2072.

Fisher, Rebecca E., et al. "Measurement of the 13C isotopic signature of methane emissions from northern European wetlands." *Global Biogeochemical Cycles* 31.3 (2017): 605-623.

Fisher, Rebecca E., et al. "Arctic methane sources: Isotopic evidence for atmospheric inputs." *Geophysical Research Letters* 38.21 (2011).

France, James L., et al. "Measurements of $\delta 13C$ in CH4 and using particle dispersion modeling to characterize sources of Arctic methane within an air mass." *Journal of Geophysical Research: Atmospheres* 121.23 (2016): 14-257.

International Energy Agency (2024) Global Methane Tracker: Methane emissions from energy. https://www.iea.org/reports/global-methane-tracker-2024/key-findings

Jacob, Daniel J., et al. "Quantifying methane emissions from the global scale down to point sources using satellite observations of atmospheric methane." *Atmospheric Chemistry and Physics* 22.14 (2022): 9617-9646.

Riddick, Stuart N., et al. "A quantitative comparison of methods used to measure smaller methane emissions typically observed from superannuated oil and gas infrastructure." *Atmospheric Measurement Techniques* 15.21 (2022): 6285-6296.

Riddick, Stuart N., et al. "Methane emissions from abandoned oil and gas wells in Colorado." *Science of The Total Environment* 922 (2024): 170990.

Zazzeri, G., et al. "Plume mapping and isotopic characterisation of anthropogenic methane sources." *Atmospheric Environment* 110 (2015): 151-162.

Zazzeri, Giulia, et al. "Carbon isotopic signature of coal-derived methane emissions to the atmosphere: from coalification to alteration." *Atmospheric Chemistry and Physics* 16.21 (2016): 13669-13680.

GLOBAL budget:

Michel, Sylvia Englund, et al. "Rapid shift in methane carbon isotopes suggests microbial emissions drove record high atmospheric methane growth in 2020–2022." *Proceedings of the National Academy of Sciences* 121.44 (2024): e2411212121.

Nisbet, Euan G., et al. "Practical paths towards quantifying and mitigating agricultural methane emissions." *Proceedings A*. Vol. 481. No. 2309. The Royal Society, 2025.

Nisbet, Euan G., et al. "Atmospheric methane: Comparison between methane's record in 2006–2022 and during glacial terminations." *Global Biogeochemical Cycles* 37.8 (2023): e2023GB007875.

Nisbet, Euan G. "New hope for methane reduction." *Science* 382.6675 (2023): 1093-1093.

Saunois, Marielle, et al. "Global methane budget 2000–2020." *Earth System Science Data Discussions* 2024 (2024): 1-147.

R: Thank you for the references you provided; we will cite some of them.

---

## Author Comment (AC3)

**Author's response for Ref: egusphere-2025-377**

**Ref. No.:** egusphere-2025-377
**Title:** Isotopic signatures of methane emission from oil and natural gas plants in southwestern China.

**Journal**: Atmospheric Chemistry and Physics.
**Authors:** Dingxi Chen, Yi Liu, Zetong Niu, Ao Wang, Pius Otwil, Yuanyuan Huang, Zhongcong Sun, Xiaobing Pang, Liyang Zhan, and Longfei Yu.

Dear Editor,

    We would like to express our sincere appreciation to you, and the anonymous reviewers for your constructive comments and suggestions on our manuscript (egusphere-2025-377) submitted to Atmospheric Chemistry and Physics. We have carefully considered all reviewers' comments and have responded to each point in detail.

Regards,
Longfei Yu
On behalf of all coauthors
May 14, 2025

- **Reviewer comments**
- Author's response

**RC2**

The paper explores the isotopic signature of the oil and gas (ONG) sector in China, one of the major sources of methane emissions. The scientific methods used in the study are sound, and the results reveal some interesting findings. I recommend major revisions to improve the clarity, and completeness of the manuscript. Given the scientific relevance of the study, it aligns well with the journal's scope and should be considered for publication, provided that all suggested revisions are implemented.

R: Thank you very much for your comments and constructive suggestions. Regarding the key aspects which require improvement, we have replied by points and will attend to the revision accordingly.

However, several aspects require improvement:

1.The manuscript requires a thorough review for grammatical errors, especially in the introduction, which needs significant rewriting. Although some errors are noted in the specific comments below, this is not an exhaustive list, and the entire manuscript would benefit from careful revision.

R: Thanks a lot for the critical comment. We see that there are still many places to revise and proof-reading. We will thoroughly review our grammatical errors with carefulness. In addition, we will ask a native speaker to help in the revision and proof-reading.

2.The source signatures of other methane sources (for example: microbial, pyrogenic) in and around this region are not considered when attributing the isotopic signature solely to ONG sources. A more comprehensive discussion of these sources would strengthen the conclusion that the primary contributor is the ONG sector.

R: Thanks for the valuable comments. As also mentioned by RC1, we agree that this is an important issue to clarify in both our method and discussion. Indeed, the location areas of these ONG sites were mostly rural, with very few residence or livestock farms. There were some scattered paddy fields and small streams, however, may not likely exert large influence on the sampling from ONG sites. Our unpublished data of $CH_4$ isotopes sampled near the paddy field of this region from a parallel study also supports our idea. In addition, either ground-based sampling points or UAV take-off points are located near middle of ONG plots or not in vicinity to the borders. Further, we considered the influence of meteorological conditions on the $CH_4$ isotope measurements, clarifying that ONG-source acts as the major drivers of the variations of isotopic signals. For more details, please refer to our earlier response to RC1 (point 3).

**3.The paper lacks a thorough discussion and comparison with previous studies. Several key papers in this field are not cited, limiting the depth and impact of the findings.**

R: Thanks for your time and effort in helping us improve the paper quality. We agree with Reviewer 2 that our work should build more link with previous studies. In the revision, we will conduct substantial revision and improvement in the introduction as well as our discussion section, to cover the current progress and gap in global ONG-$CH_4$ isotope research. Further, we will also discuss the uncertainty of the study, as well as the value for future work in evaluation $CH_4$ leakage from Chinese ONGs and global anthropogenic CH4 budgets.

**A few specific comments:**

- **There should be space before the references in the text.**

R: Thank you for your suggestion. We will add spaces before the references in the text.

- **There should be space between the numbers and units (for example - 45.06 ‰).**

R: Thank you for your suggestion. We will add spaces between numbers and units.

- **Adding the measurements from this study to the general source composition figure would enhance visualization and provide a better context within a global framework.**

R: Thank you for your constructive suggestion. We will include this content in the revised manuscript.

**L20: to the atmosphere from the Chinese oil and gas**

R: Thanks. We will add "the" in the revised manuscript.

**L63: account**

R: Thanks. We will change "accounting" to "account".

**L66-L67: This sentence is not clear.**

R: Thanks. We will revise it.

**L69-L70: This sentence is not clear.**

R: Thanks. We will revise it.

**L78: isotopic composition?**

R: Thanks. We will revise it.

**L83-L85: Grammatically incorrect**

R: Thanks. We will revise it.

**L92: research**

R: Thanks. We will change "researches" to "research".

**L120-121: Grammatically incorrect**

R: Thanks. We will revise it.

**L177: bags were lifted to the altitude**

R: Thanks. We will change "was" to "were".

**L183: sentence unclear**

R: Thanks. We will revise it.

**L220: rewrite this sentence**

R: Thanks. We will rewrite this sentence.

**L227: 'method of' is a repetition**

R: Thanks. We will delete "method of".

**L378: the Keeling plot**

R: Thanks. We will change "keeling plot" to "the Keeling plot".

**L479 and L512: wrong reference formats**

R: Thanks. We will revise it.

---

## Author Response (AR3)

**Review 1:**

In your response to my first general review point you cite a range of references as having used data from Picarro isotopic instruments. Please check where you have cited these in the manuscript and that what you are saying is correct, because Menoud et al. (2022) focusses on samples measured by IRMS (the isotope Picarro used on ground surveys was mostly for the mole fraction measurements, not for the isotopic signatures). Additionally al-Shalan et al. (2022) did not make any measurements on an isotope Picarro and all samples were collected in bags for laboratory IRMS analysis. The Lu et al (2022) paper mentions that an isotopic Picarro was used temporaility due to failure of other instruments, mostly for mole fraction measurements, but the isotopic interpretation is based around the many bag samples that were sent to 2 IRMS labs for analysis. Therefore only 3 of the 5 papers focus on the use and interpretation of isotope Picarro measurements.

R: Many thanks for the reviewer to point out this imprecise description. We have now checked these citations in the final revision and have additionally referred our Picarro measurements to the work conducted with Picarro instruments (Line 201-203 in TC version).

Please check the new sections again, as there are a few places where sentencees do not make complete sense the ways they are currently written.

R: Thanks for the suggestion. We have carefully gone through the whole manuscript again.

**Reviewer 2:**

The paper has been extensively revised and is now much more complete. As it contains a great deal of very useful isotopic information for regional and global studies it should be encouraged to publication.

Line 512/513 should probably focus primarily on Michel et al 2024, as should line 579, rather than prioritising these older papers.

Overall, the paper is much better and should be published.

R: We appreciate the reviewer's suggestion. As for Line 512-513, we have updated the new citation (Line 518 in TC version).